# Isotropic Noise in Stochastic and Quantum Convex Optimization

Annie Marsden[*]      Liam O'Carroll[†]      Aaron Sidford[†]      Chenyi Zhang[†]

## Abstract

We consider the problem of minimizing a $d$-dimensional Lipschitz convex function using a stochastic gradient oracle. We introduce and motivate a setting where the noise of the stochastic gradient is *isotropic* in that it is bounded in every direction with high probability. We then develop an algorithm for this setting which improves upon prior results by a factor of $d$ in certain regimes, and as a corollary, achieves a new state-of-the-art complexity for sub-exponential noise. We give matching lower bounds (up to polylogarithmic factors) for both results. Additionally, we develop an efficient *quantum isotropifier*, a quantum algorithm which converts a variance-bounded quantum sampling oracle into one that outputs an unbiased estimate with isotropic error. Combining our results, we obtain improved dimension-dependent rates for quantum stochastic convex optimization.

## 1 Introduction

Stochastic convex optimization (SCO) [51, 47, 48, 9] is a foundational problem in optimization and learning theory, with numerous applications in theoretical computer science [16], operations research [50], machine learning [11], and beyond. Algorithms for SCO, most notably stochastic gradient descent (SGD) and its many variants, are extensively studied and widely deployed in practice. In addition, there has been increasing interest recently in its quantum analog, *quantum stochastic convex optimization*, and new provably efficient quantum algorithms with have been developed [53, 67].

In one of its simplest forms which we focus on in this paper, SCO asks to compute an $\epsilon$-optimal point of a differentiable[3] convex $L$-Lipschitz function $f : \mathbb{R}^d \to \mathbb{R}$, minimized at an unknown point $x^\star$ with $\|x^\star\| \leq R$ for the Euclidean norm $\|\cdot\|$, given access to a bounded stochastic gradient oracle, which we define formally in Definition 1 and Definition 2 below. Throughout, we assume that the randomness in calls to any oracle are independent of previous calls.

**Definition 1** (SGO). $\mathcal{O}(\cdot)$ *is a* stochastic gradient oracle (SGO) *for differentiable* $f : \mathbb{R}^d \to \mathbb{R}$ *if when queried at* $x \in \mathbb{R}^d$, *it outputs* $\mathcal{O}(x) \in \mathbb{R}^d$ *such that* $\mathbb{E}\mathcal{O}(x) = \nabla f(x)$.

**Definition 2** (BSGO). $\mathcal{O}_B(\cdot)$ *is a* $\sigma_B$-bounded SGO ($\sigma_B$-BSGO) *for differentiable* $f : \mathbb{R}^d \to \mathbb{R}$ *if it is an SGO for* $f$ *such that* $\mathbb{E}\|\mathcal{O}_B(x)\|^2 \leq \sigma_B^2$ *for any query* $x \in \mathbb{R}^d$.

It is well known that SGD, in particular, iterating $x_{t+1} \leftarrow x_t - \eta\mathcal{O}_B(x_t)$ with an appropriate choice of step size $\eta$, solves SCO with $O(R^2\sigma_B^2/\epsilon^2)$ queries, which is optimal even when $d = 1$ [2]. In this paper, we seek to bypass this fundamental limit by studying more fine-grained models of the stochastic gradient. In particular, we study the following question:

---

[*]Google Deepmind, anniemarsden@google.com

[†]Stanford University, {ocarroll,sidford,chenyiz}@stanford.edu

 Preprint available at arXiv:2510.20745.

[3]We assume all objective functions are differentiable. Following a similar argument in prior works (e.g., [17, 53]), our results extend to non-differentiable settings because convex functions are almost everywhere differentiable, and our algorithms are robust to polynomially small numerical errors.

39th Conference on Neural Information Processing Systems (NeurIPS 2025).

*Can we identify shape-dependent assumptions on the noise of the stochastic gradient which allow us to obtain new state-of-the-art algorithmic guarantees?*

One of the most natural shape-dependent assumptions is to assume a bound on the variance of the SGO, which we capture in the following definition:

**Definition 3** (VSGO). $\mathcal{O}_V(\cdot)$ *is a* $\sigma_V$-*variance-bounded SGO* ($\sigma_V$-*VSGO*) *for differentiable* $f :$ $\mathbb{R}^d \to \mathbb{R}$ *if it is an SGO for* $f$ *such that* $\mathbb{E}\|\mathcal{O}_V(x) - \nabla f(x)\|^2 \leq \sigma_V^2$ *for any query* $x \in \mathbb{R}^d$.

In the setting we consider where $f$ is $L$-Lipschitz (see Problem 1 for a formal definition), SGD trivially achieves a rate of $O(R^2(L^2 + \sigma_V^2)/\epsilon^2) = O(R^2\sigma_V^2/\epsilon^2 + R^2L^2/\epsilon^2)$ under a $\sigma_V$-VSGO due to the observation that a $\sigma_V$-VSGO is a $O(L + \sigma_V)$-BSGO. Inspecting the above rate, the $R^2\sigma_V^2/\epsilon^2$ term is unimprovable even when $d = 1$ [2].[4] However, the same is not true for the $R^2L^2/\epsilon^2$ term, which stems from the non-stochastic component of the problem. Indeed, when $\sigma_V = 0$, in which case the SGO is a gradient oracle, the $O(R^2L^2/\epsilon^2)$ rate (which is achieved by gradient descent [49]) is optimal only for $\epsilon \geq \Omega(RL/\sqrt{d})$ [24]; cutting plane methods which achieve $\tilde{O}(d)$ rates are better at smaller target precisions, where we use $\tilde{O}(\cdot)$ throughout the paper to hide polylogarithmic factors in $1/\epsilon$, $\log(1/\delta)$, $d$, $R$, and $L$.

This leaves the door open for improved *dimension-dependent* rates for SCO under a VSGO, which have not been explicitly studied to the best of our knowledge. In particular, in light of the fact that a $O(R^2\sigma_V^2/\epsilon^2 + R^2L^2/\epsilon^2)$ rate is achievable by SGD and it is possible to achieve $\min\{O(R^2L^2/\epsilon^2), \tilde{O}(d)\}$ when $\sigma_V = 0$, this begs the natural open problem:

**Open Problem 1.** *Is it possible to solve SCO with* $\tilde{O}(R^2\sigma_V^2/\epsilon^2 + d)$ *queries to a* $\sigma_V$-*VSGO?*

While we do not solve this open problem, we nonetheless make substantial progress in characterizing the complexity of SCO under more fine-grained shape-dependent assumptions on the SGO. Our main conceptual contribution is to answer this open problem in the affirmative under a stronger noise model which we introduce in the next section, termed *isotropic noise*. We provide a dimension-dependent algorithm in this setting which achieves the conjectured $\tilde{O}(R^2\sigma^2/\epsilon^2 + d)$ rate (albeit $\sigma^2$ is not the variance but a different parameter associated with our noise model). We then instantiate this result to achieve the target $\tilde{O}(R^2\sigma^2/\epsilon^2 + d)$ rate for sub-exponential noise, a widely studied class of distributions in machine learning and statistics, as well as a $\tilde{O}(dR^2\sigma_V^2/\epsilon^2 + d)$ rate for SCO under a VSGO,[5] which is worse than the conjectured rate by a $d$-factor. We then give lower bounds for isotropic noise and sub-exponential noise which show that in any parameter regime, applying the better of our algorithm and SGD is optimal (up to polylogarithmic factors).

Finally, we leverage our algorithm to obtain a new state-of-the-art guarantee for quantum SCO under a quantum analog of the VSGO oracle. In what may be of independent interest, our quantum result relies on a quantum subroutine which we call a *quantum isotropifier*, which converts a quantum analog of the VSGO oracle into one which outputs an unbiased estimate of the gradient with isotropic noise. In other words, this subroutine allows us to apply our improved guarantees for isotropic stochastic gradients to an even broader class of noise models in the quantum setting.

In the remainder of the introduction, we define our noise model and present our results in more detail (Section 1.1), discuss additional related work (Section 1.2), define notation (Section 1.3), and give a roadmap of the rest of the paper (Section 1.4).

## 1.1   Results

**Isotropic and sub-exponential SGOs.**   As our main conceptual contribution, we define a new noise model for SCO which captures the impact of the *shape* of the noise, as opposed to only its moments. In particular, we consider an *isotropic noise model*, formalized in the following definition, where the magnitude of the noise in any fixed direction is bounded with probability at least $1 - \delta$.

---

[4]This follows from the same lower bound construction which shows that $\Omega(R^2\sigma_B^2/\epsilon^2)$ queries are necessary for SCO under a $\sigma_B$-BSGO. Indeed, in that construction (e.g., Section 5 in [23]), it is the case that $\sigma_V^2 = \Theta(\sigma_B^2)$.

[5]We emphasize that although they study a different setting, this rate can be recovered by modifying the techniques of [53]; see Sections 1.1 and 2 for further discussion.

**Definition 4** (ISGO). $\mathcal{O}_I(\cdot)$ *is a $(\sigma_I, \delta)$-isotropic SGO ($(\sigma_I, \delta)$-ISGO) for differentiable $f : \mathbb{R}^d \to \mathbb{R}$ if it is an SGO for $f$ such that*

$$\mathbb{P}\Big[|\langle \mathcal{O}_I(x) - \nabla f(x), u\rangle| \geq \sigma_I/\sqrt{d}\Big] \leq \delta \ \text{for any } x, u \in \mathbb{R}^d \ \text{s.t.} \ \|u\| = 1. \tag{1}$$

The $1/\sqrt{d}$ scaling factor in Eq. 1 is included to ensure $\sigma_I$ in Definition 4 and $\sigma_V$ in Definition 3 are comparable in scale. For example, a $(\sigma, 0)$-ISGO $\mathcal{O}_I(\cdot)$ is also a $\sigma$-VSGO, since with $e_i$ as the $i$-th standard basis vector:

$$\mathbb{E}\|\mathcal{O}_I(x) - \nabla f(x)\|^2 = \mathbb{E}\sum_{i\in[d]}\langle e_i, \mathcal{O}_I(x) - \nabla f(x)\rangle^2 = \sum_{i\in[d]}\mathbb{E}\langle e_i, \mathcal{O}_I(x) - \nabla f(x)\rangle^2 \leq \sigma^2.$$

Informally, an ISGO can be thought of as a VSGO with the stronger assumption that the noise is bounded evenly in all directions, thereby imposing additional structure on its *shape*. Formally, we show later in Lemma 9 that a $(6\sigma_V\sqrt{d}, \delta)$-ISGO query can be implemented with logarithmically many queries to a $\sigma_V$-VSGO, thereby allowing us to translate algorithmic guarantees for an ISGO to a VSGO at the cost of a $\sqrt{d}$-factor.

ISGOs also naturally capture a variety of light-tailed distributions, for which we use sub-exponential distributions as a representative example throughout this paper due to their widespread prevalence in machine learning and statistics. In particular, we consider the following ESGO:

**Definition 5** (ESGO). $\mathcal{O}_E(\cdot)$ *is a $\sigma_E$-sub-exponential SGO ($\sigma_E$-ESGO) for differentiable $f : \mathbb{R}^d \to \mathbb{R}$ if it is an SGO for $f$ such that for any $x, u \in \mathbb{R}^d$ such that $\|u\| = 1$:*

$$\mathbb{P}[|\langle \mathcal{O}_E(x) - \nabla f(x), u\rangle| \geq t] \leq 2 \cdot \exp(-t\sqrt{d}/\sigma_E) \ \text{for all } t \geq 0. \tag{2}$$

The $\sqrt{d}$ factor in Definition 5 is present for the same reason it is present Definition 4, and we show for all $\delta \in (0, 1)$ that any $\sigma$-ESGO is a $(\sigma\log(2/\delta), \delta)$-ISGO (see Lemma 8). We collect and review basic facts about sub-exponential distributions, as well as provide some additional discussion about the relationships between the various SGO oracles we define, in Appendix D. In particular, we show that a $\sigma_E$-ESGO is a $C\sigma_E$-VSGO for some absolute constant $C$, and thus Definition 5 can also be interpreted as imposing structure on the shape of the noise beyond a variance bound.

**The complexity of SCO with ISGOs, ESGOs, and VSGOs.**   We formalize the general setting we consider in Problem 1 and then present our results, including upper and lower bounds.

**Problem 1** (SCO). *Given access to an SGO $\mathcal{O}(\cdot)$ for a convex and $L$-Lipschitz (with respect to the $\ell_2$-norm) function $f : \mathbb{R}^d \to \mathbb{R}$ such that there exists $x^\star \in \operatorname{argmin}_{x\in\mathbb{R}^d} f(x)$ with $\|x^\star\| \leq R$, the goal is to output an $\epsilon$-optimal point.*

In Theorem 1 and Corollary 2 (proven in Section A and Section B respectively), we achieve upper and lower bounds for Problem 1 given access to a $(\sigma_I, \delta)$-ISGO. We obtain the upper bound Theorem 1 via a *stochastic cutting plane method* by extending the techniques of [53], and the lower bound Corollary 2 by reducing a mean estimation problem to Problem 1; see Section 2 for a technical overview. In Corollary 2, we use $\tilde{\Omega}(\cdot)$ to hide logarithmic factors in $d$.

**Theorem 1.** *Problem 1 can be solved with probability at least $2/3$ in $\tilde{O}\big(R^2\sigma_I^2/\epsilon^2 + d\big)$ queries to a $(\sigma_I, \delta)$-ISGO for any $\delta \leq \frac{1}{Md(R^2\sigma_I^2/\epsilon^2+d)}$, where $M = \tilde{O}(1)$.*

**Corollary 2.** *Any algorithm which solves Problem 1 with probability at least $2/3$ using a $(\sigma_I, \delta)$-ISGO makes at least $\tilde{\Omega}(R^2\sigma_I^2/(\epsilon^2\log^2(1/\delta)) + \min\{R^2L^2/\epsilon^2, d\})$ queries.*

In other words, Theorem 1 says that an $\tilde{O}(R^2\sigma_I^2/\epsilon^2 + d)$ rate is achievable as long as $\delta$ is inverse-polynomially small, and Corollary 2 says this is tight up to polylog factors when the target precision is sufficiently small so that $d \leq R^2L^2/\epsilon^2$ or equivalently $\epsilon \leq RL/\sqrt{d}$. In particular, Theorem 1 demonstrates that shape-dependent assumptions on the noise can lead to speedups over the known rates for SGD. For example, for a $(\sigma, 0)$-ISGO (which we saw above is also a $\sigma$-VSGO), the $\tilde{O}\big(R^2\sigma^2/\epsilon^2 + d\big)$ rate of Theorem 1 improves upon the $O(R^2\sigma^2/\epsilon^2 + R^2L^2/\epsilon^2)$ rate of SGD when $L \gg \max\{\sigma, \epsilon\sqrt{d}/R\}$, which captures a variety of high precision, low variance regimes.

Next, using the fact discussed above that ISGO queries can be implemented via ESGO and VSGO oracles, we obtain the following upper bounds (proven in Section A) as corollaries of Theorem 1:

**Corollary 3.** *Problem 1 can be solved with probability at least $2/3$ in $\tilde{O}(R^2\sigma_E^2/\epsilon^2 + d)$ queries to a $\sigma_E$-ESGO.*

**Corollary 4.** *Problem 1 can be solved with probability at least $2/3$ in $\tilde{O}(dR^2\sigma_V^2/\epsilon^2 + d)$ queries to a $\sigma_V$-VSGO.*

Thus, sub-exponential noise allows us to match the rate conjectured in Open Problem 1, whereas using a VSGO results in an additional $d$ factor in the variance-dependent term. Nevertheless, Corollary 4 still outperforms the $O(R^2\sigma_V^2/\epsilon^2 + R^2L^2/\epsilon^2)$ rate achieved by SGD for solving Problem 1 given a $\sigma_V$-VSGO when $L \gg \sqrt{d} \cdot \max\{\sigma_V, \epsilon/R\}$. We emphasize that while they study a different setting, a modification of the techniques of [53] can achieve the same complexity as Corollary 4; see the technical overview in Section 2 for further discussion. However, we believe the fact that we are able to recover Corollary 4 as a simple consequence using our more general $(\sigma_I, \delta)$-ISGO primitive illustrates its potential broader applicability.

Finally, we prove in Section B that Corollary 3 is tight up to polylog factors when $d \geq R^2L^2/\epsilon^2$:

**Theorem 5.** *Any algorithm which solves Problem 1 with probability at least $2/3$ using a $\sigma_E$-ESGO makes at least $\tilde{\Omega}(R^2\sigma_E^2/\epsilon^2 + \min\{R^2L^2/\epsilon^2, d\})$ queries.*

This effectively solves Open Problem 1 for the special case of sub-exponential noise in light of the fact that SGD matches our lower bound in the other regime where $d \leq R^2L^2/\epsilon^2$.

**Quantum SCO under variance-bounded noise.** As an additional application of Theorem 1, we consider stochastic convex optimization (SCO) with access to a quantum analog of a VSGO, which we refer to as a QVSGO. The QVSGO is a direct and natural extension of a VSGO, and slightly generalizes the notion of QSGO introduced in [53, Definition 3].

**Definition 6** (QVSGO). *For $f\colon \mathbb{R}^d \to \mathbb{R}$ and $\sigma_V > 0$, its quantum bounded stochastic gradient oracle ($\sigma_V$-QVSGO) is defined as*[6][7]

$$\mathcal{O}_{QV}\,|x\rangle \otimes |0\rangle \to |x\rangle \otimes \int_{g\in\mathbb{R}^d} \sqrt{p_{f,x}(g)\mathrm{d}g}\,|g\rangle \otimes |\mathrm{garbage}(g)\rangle, \tag{3}$$

*where $p_{f,x}(\cdot)$ is the probability density function of the stochastic gradient that satisfies*

$$\mathop{\mathbb{E}}_{g\sim p_{f,x}}[g] = \nabla f(x), \qquad \mathop{\mathbb{E}}_{g\sim p_{f,x}}\|g - \nabla f(x)\|^2 \leq \sigma_V^2.$$

Given access to a QVSGO, we develop a quantum algorithm that achieves improved query complexity for the following quantum SCO problem. This problem slightly generalizes [53, Problem 1] by introducing an additional parameter $\sigma_V$ alongside the Lipschitzness $L$.

**Problem 2** (Quantum SCO). *Given query access to a $\sigma_V$-QVSGO $\mathcal{O}_{QV}$ for a convex and $L$-Lipschitz (with respect to the $\ell_2$-norm) function $f\colon \mathbb{R}^d \to \mathbb{R}$ such that there exists $x^\star \in \mathrm{argmin}_{x\in\mathbb{R}^d} f(x)$ with $\|x^\star\| \leq R$, the goal is to output an $\epsilon$-optimal point.*

To solve Problem 2, we develop an unbiased quantum multivariate mean estimation algorithm whose error is isotropic, which we refer to as *quantum isotropifier*. We use this algorithm to prepare an ISGO using a QVSGO:

**Theorem 6.** *For any differentiable $f\colon \mathbb{R}^d \to \mathbb{R}$, a $(\sigma_I, \delta)$-ISGO of $f$ can be implemented using $\tilde{O}(\sigma_V\sqrt{d}\log^7(1/\delta)/\sigma_I)$ queries to a $\sigma_V$-QVSGO.*

Combining Theorem 1 and Theorem 6, we obtain the following improved query complexity for quantum SCO (proven in Section C):

**Theorem 7.** *With success probability at least $2/3$, Problem 2 can be solved using $\tilde{O}(dR\sigma_V/\epsilon)$ queries to a $\sigma_V$-QVSGO.*

---

[6]Considering such quantum extensions of classical oracles is standard in the literature, see, e.g., [20, 65, 53]. Moreover, theoretically there are well-established techniques for implementing these quantum analogs of classical oracles. Specifically, if a classical oracle can be implemented via a classical circuit, its corresponding quantum oracle can be implemented using a quantum circuit of the same size.

[7]A description of the quantum notation we use can be found in Section C.

To compare, the prior state-of-the-art for solving Problem 2 includes two quantum algorithms of [53] which achieve query complexities of $\tilde{O}(d^{3/2}(L + \sigma_V)R/\epsilon)$ and $\tilde{O}(d^{5/8}((L + \sigma_V)R/\epsilon)^{3/2})$. With minor modifications, the query complexity of the first algorithm can be reduced to $\tilde{O}(d^{3/2}\sigma_V R/\epsilon)$. However, it remains unclear whether a similar reduction in query complexity can be achieved for the second algorithm. In comparison, our algorithm achieves a polynomial improvement in $d$.

## 1.2 Additional related work

**Stochastic convex optimization.** SCO is a foundational problem in optimization theory and has been extensively studied for decades. In particular, SCO with a VSGO has been studied under different assumptions on the objective function $f$ than the $L$-Lipschitz assumption we focus on in this paper. Notably, if the gradient of $f$ is $\beta$-Lipschitz, the seminal paper [39] achieved an optimal error of $O(\beta/T^2 + \sigma_V/\sqrt{T})$ after $T$ queries. We note that our guarantees for Problem 1 can be extended to the setting where the gradient of $f$ is $\beta$-Lipschitz at the cost of only polylogarithmic factors since our bounds have polylogarithmic dependence on the Lipschitz constant of $f$. This follows because a function with a $\beta$-Lipschitz gradient which is minimized in a ball of radius $R$ is itself $O(\beta R)$-Lipschitz in the ball.

Beyond the bounded variance assumption, [62] examines SGOs with heavy-tailed or infinite variance noise and shows that stochastic mirror descent is optimal in such cases. As for cutting plane methods, there is a long line of work in deterministic settings which has sought to improve the query complexity and/or runtime; see e.g. [56, 40, 34].

**Quantum mean estimation.** Quantum mean estimation and the closely closely related problems of amplitude estimation and phase estimation have been extensively studied [1, 54, 32, 64, 14, 46, 31]. The seminal amplitude estimation algorithm [15] can be used to obtain a quadratic improvement on the query complexity for estimating the mean of any Bernoulli random variable. Building upon this result, [30] (whose query complexity is further improved by [38]) introduced a quantum sub-Gaussian mean estimator that achieves a quadratically better query complexity than the classical counterpart while providing a mean estimate with sub-Gaussian error, without requiring additional assumptions on the variance or tail behavior of the underlying distribution. Multilevel Monte Carlo methods have also been widely used in quantum algorithms [3, 42].

**Quantum algorithms and lower bounds for optimization problems.** There has been a rich study of quantum algorithms for classical optimization problems, including semidefinite programs [7, 13, 12, 37, 58, 60], convex optimization [6, 59, 19, 41], and non-convex optimization [65, 21, 28, 43, 41]. Despite these recent progress in quantum algorithms using quantum evaluation oracles and quantum gradient oracles, their limitations have also been explored in a series of works establishing quantum lower bounds for certain settings of convex optimization [25, 66] and non-convex optimization [66].

## 1.3 Notation

$\|\cdot\|$ is the $\ell_2$-norm. We define $B_p(r, x_0) := \{x \in \mathbb{R}^d : \|x - x_0\|_p \leq r\}$ to be the $\ell_p$-ball of radius $r$ centered at $x_0$, and use $B_p(r) := B_p(r, 0)$ and $B_p := B_p(1, 0)$. A halfspace is denoted by $H_\geq(a \in \mathbb{R}^d, b \in \mathbb{R}) := \{x \in \mathbb{R}^d : a^\top x \geq b\}$. For $f : \mathbb{R}^d \to \mathbb{R}$, we let $f^\star := \inf_x f(x)$ and call $x \in \mathbb{R}^d$ $\epsilon$-*(sub)optimal* if $f(x) \leq f^\star + \epsilon$. A function $f : \mathbb{R}^d \to \mathbb{R}$ is $L$-*Lipschitz* if $f(x) - f(y) \leq L\|x - y\|$ for all $x, y \in \mathbb{R}^d$. For two matrices $A, B \in \mathbb{R}^{d \times d}$, we write $A \preceq B$ if $x^\top A x \leq x^\top B x$ for all $x \in \mathbb{R}^d$. The standard basis in $\mathbb{R}^d$ is denoted $\{e_1, \ldots, e_d\}$. For random variables $X, Y$, we use $H(X) := -\sum_x p(x) \log p(x)$ to denote the entropy of $X$, use $H(Y|X) := H(X, Y) - H(X)$ to denote the conditional entropy of $Y$ with respect to $X$, and use $I(X; Y) := H(X) + H(Y) - H(X, Y)$ to denote the mutual information between $X$ and $Y$. The cardinality of a finite set $S$ is denoted $|S|$. We use $\tilde{O}(\cdot)$ and $\tilde{\Omega}(\cdot)$ to denote big-$O$ notation omitting polylogarithmic factors.

## 1.4 Paper organization

We give a technical overview of the paper in Section 2 and conclude in Section 3. We prove our upper bounds for Problem 1 under ISGO, ESGO, and VSGO oracles (resp. Theorem 1 and Corollaries 3 and 4) in Appendix A. We establish our lower bounds for Problem 1 given either an ISGO or ESGO

oracle in Appendix B. In Appendix C, we present our *quantum isotropifier* and then apply it to obtain an improved bound for quantum SCO under variance-bounded noise. In Appendix D, we review sub-exponential distributions and further discuss the relationships between the various stochastic gradient oracles we study. Finally, we collect miscellaneous technical lemmas in Appendix E.

## 2 Technical overview

In this section, we present high-level overviews of our techniques. In Section 2.1, we give an overview of our *stochastic cutting plane method*, based on an extension of the techniques of [53], which we use to obtain our upper bounds for Problem 1. In Section 2.2, we give a technical overview of our lower bounds for Problem 1. We conclude in Section 2.3 with a discussion of our application to quantum SCO; we believe our *quantum isotropifier* subroutine discussed in that section may be of independent interest. The formal proofs of the results discussed in Sections 2.1, 2.2, and 2.3 can be found in Sections A, B, and C respectively.

### 2.1 Overview of our upper bounds for SCO

Recall that our main non-quantum algorithmic result is Theorem 1, which says that $\tilde{O}(R^2\sigma_I^2/\epsilon^2 + d)$ queries to a $(\sigma_I, \delta)$-ISGO suffice to solve Problem 1 when $\delta$ is inverse-polynomially small. We prove this upper bound using a *stochastic cutting plane method* via an extension of the techniques of [53] (see Section A for formal proofs). Cutting plane methods are a foundational class of algorithms in convex optimization which solve *feasibility problems,* for which we use the following formulation: (recall $H_\geq(a \in \mathbb{R}^d, b \in R)$ denotes the halfspace $\{x \in \mathbb{R}^d : a^\top x \geq b\}$):

**Definition 7** (Feasibility Problem). *For $R' \geq r > 0$, we define the $(R', r)$-feasibility problem as follows: We are given query access to a (potentially randomized)* halfspace oracle *which when queried at $x \in B_2(R')$, outputs a vector $g_x \in \mathbb{R}^d \setminus \{0\}$. The goal is to query the oracle at a sequence of points $x_1, \ldots, x_T \in B_2(R')$ such that $B_2(R') \bigcap_{t \in [T]} H_\geq(g_{x_t}, g_{x_t}^\top x_t)$ does not contain a ball of radius $r$ (namely, any set of the form $B_2(r, z)$ for $z \in \mathbb{R}^d$). We call an algorithm a $\mathcal{T}$-algorithm for the $(R', r)$-feasibility problem if it achieves this goal with at most $\mathcal{T}$ queries.*

A variety of cutting plane methods [56, 40, 34] are $\tilde{O}(d)$-algorithms for the $(R', r)$-feasibility problem, where $\tilde{O}(\cdot)$ hides logarithmic factors in $d$, $R'$, and $1/r$. Furthermore, it is well known that solving the feasibility problem where the halfspace oracle is given by the negative gradient of $f$ suffices to solve Problem 1 using $\tilde{O}(d)$ queries to an exact gradient oracle, where $\tilde{O}(\cdot)$ hides logarithmic factors in $R$, $d$, $L$, and $1/\epsilon$. (Technically, there is also an additional post-processing step needed which we discuss later.) This reduction follows from the fact that by the Lipschitzness of $f$, every point in the ball $B_2(\epsilon/L, x^\star) := \{z \in \mathbb{R}^d : \|z - x^\star\| \leq \epsilon/L\}$ is $\epsilon$-optimal. Therefore, setting $r := \epsilon/L$ and $R' := 2R$ so that $B_2(\epsilon/L, x^\star) \subseteq B_2(R')$ (we can assume $r \leq R$ without loss of generality since if $\epsilon > RL$, the origin is $\epsilon$-optimal) and running one of the aforementioned $\tilde{O}(d)$-algorithms for the $(R', r)$-feasibility problem with the halfspace oracle given by $-\nabla f(\cdot)$, we must have queried the oracle at an iterate $x_t$ such that $-\nabla f(x_t)^\top z < -\nabla f(x_t)^\top x_t$ for some $z \in B_2(\epsilon/L, x^\star)$. This implies $x_t$ is $\epsilon$-optimal by convexity:

$$f(x_t) \leq f(z) + \langle \nabla f(x_t), x_t - z \rangle \leq f(z) \leq f(x^\star) + \epsilon.$$

Recently, [53], which broadly studies quantum algorithms for stochastic optimization in a variety of settings, observed the above analysis can be extended to the setting where we only have access to an *approximate gradient,* which they capture as follows (paraphrased from Definition 5 in [53]):

**Definition 8** (AGO). *$\tilde{h}(\cdot)$ is a $\gamma$-approximate gradient oracle ($\gamma$-AGO) if when queried at $x \in \mathbb{R}^d$, the (potentially random) output $\tilde{h}(x) \in \mathbb{R}^d$ satisfies $\|\tilde{h}(x) - \nabla f(x)\| \leq \gamma$.*

In other words, a $\gamma$-AGO $\tilde{h}(\cdot)$ gives an estimate of the gradient with at most $\gamma$ error in $\ell_2$-norm. Then running an $\tilde{O}(d)$-algorithm for the $(R' := 2R, r := \epsilon/L)$-feasibility problem as before, except with $-\tilde{h}(\cdot)$ as the halfspace oracle as opposed to $-\nabla f(x)$, implies there exists an iterate $x_t$ and $\epsilon$-optimal $z \in B_2(\epsilon/L, x^\star)$ such that $-h_t^\top z < -h_t^\top x_t$, where $h_t$ was the result of calling $\tilde{h}(\cdot)$ with input $x_t$ at

the $t$-th step. Thus, by convexity:

$$
\begin{aligned}
f(x_t) &\leq f(z) + \langle \nabla f(x_t), x_t - z \rangle \\
&= f(z) + \underbrace{\langle h_t, x_t - z \rangle}_{\text{①}} + \underbrace{\langle \nabla f(x_t) - h_t, x_t - z \rangle}_{\text{②}} \leq f(x^\star) + \epsilon + 4R\gamma,
\end{aligned}
$$

where the last inequality follows because $z$ is $\epsilon$-optimal; ① $< 0$ by assumption; and ② $\leq \|\nabla f(x_t) - h_t\|\|x_t - z\| \leq 4R\gamma$ by the Cauchy-Schwarz inequality, Definition 8, and the fact that $x_t, z \in B_2(R') = B_2(2R)$. In particular, if $\gamma \leq \epsilon/(4R)$, then $x_t$ is $2\epsilon$-optimal.

[53] used this observation to obtain then state-of-the-art quantum algorithms for Problem 1 using a quantum analog of the BSGO oracle (see Definition 2). In particular, they implement an $\epsilon/(4R)$-AGO at each step with high probability using a quantum variance reduction procedure. While it is not explicitly stated, this analysis can be extended to the non-quantum setting by mini-batching. One can show that an $\epsilon/(4R)$-AGO query can be implemented using $\tilde{O}(R^2\sigma_V^2/\epsilon^2 + 1)$ queries to a $\sigma_V$-VSGO $\mathcal{O}_V(\cdot)$ with success probability at least $1 - \xi$, where $\tilde{O}(\cdot)$ hides logarithmic factors in $1/\xi$. Then choosing $\xi$ appropriately to union bound the failure probabilities over all iterations, an $\tilde{O}(d)$-algorithm for the feasibility problem with the halfspace oracle given by this mini-batching procedure yields an $\tilde{O}(dR\sigma_V^2/\epsilon^2 + d)$ rate for solving Problem 1 given a $\sigma_V$-VSGO, matching the rate we recover in Corollary 4 as a result of our more general framework.

Our improvements over [53] are built on the key observation that it in fact suffices to only *weakly* control the error of the approximate gradient in $\ell_2$-norm (it can be polynomially large), as long as we *tightly* control the error of the approximate gradient in a particular fixed direction—namely, the direction to the optimum $x^\star$. To start, we refine Definition 8 as follows by defining a *marginal approximate gradient oracle*:

**Definition 9** (MAGO). $\tilde{g}(\cdot, \cdot)$ *is a* marginal $(\eta \geq 0, \Gamma \geq 0)$-approximate gradient oracle *(*$(\eta, \Gamma)$-MAGO) if when queried at* $x, u \in \mathbb{R}^d$, *the (potentially random) output* $\tilde{g}(x, u) \in \mathbb{R}^d$ *satisfies*

$$
\|\tilde{g}(x, u) - \nabla f(x)\| \leq \Gamma \ \text{ and } \ |\langle \tilde{g}(x, u) - \nabla f(x), \tilde{u} \rangle| \leq \eta \ \text{ for } \tilde{u} := u/\|u\|. \tag{4}
$$

Next, consider running a cutting plane method for the $(R' := 2R, r := \min\{\epsilon/L, \epsilon/\Gamma\})$-feasibility problem, where the halfspace oracle at a query point $x \in \mathbb{R}^d$ is given by $-\tilde{g}(x, x - x^\star)$. As before, there exists an iterate $x_t$ and $\epsilon$-optimal $z \in B_2(r, x^\star)$ such that $-g_t^\top z < -g_t^\top x_t$, where $g_t$ was the result of calling $\tilde{g}(x_t, x_t - x^\star)$ at the $t$-th step. Then by convexity:

$$
\begin{aligned}
f(x_t) &\leq f(z) + \langle \nabla f(x_t), x_t - z \rangle \\
&= f(z) + \underbrace{\langle g_t, x_t - z \rangle}_{\text{③}} + \underbrace{\langle \nabla f(x_t) - g_t, x_t - x^\star \rangle}_{\text{④}} + \underbrace{\langle \nabla f(x_t) - g_t, x^\star - z \rangle}_{\text{⑤}} \\
&\leq f(x^\star) + 2\epsilon + 4R\eta,
\end{aligned}
$$

where the last inequality followed because $z$ is $\epsilon$-optimal; ③ $< 0$ by assumption; by Definition 9 and a triangle inequality ④ $\leq \eta\|x_t - x^\star\| \leq 4R\eta$; and finally ⑤ $\leq \|\nabla f(x_t) - g_t\|\|x^\star - z\| \leq \Gamma r \leq \epsilon$ by Cauchy-Schwarz, Definition 9, and the choice of $r$. Thus, if $\eta \leq \epsilon/(4R)$, then $x_t$ is $3\epsilon$-optimal. Crucially, because cutting plane methods for the feasibility problem have logarithmic dependence on $1/r$, a polynomial bound on $\Gamma$ suffices to maintain an $\tilde{O}(d)$-query complexity for this problem.

Next, we show how to implement a MAGO using a $(\sigma_I, \delta)$-ISGO in the following lemma, proven in Section A:

**Lemma 1.** *For any* $\delta, \xi \in (0, 1)$; $x, u \in \mathbb{R}^d$; *and* $\eta > 0$, *a query* $\tilde{g}(x, u)$ *to an* $(\eta, \Gamma := \eta\sqrt{d})$-MAGO *can be implemented using* $K = O(\frac{\sigma_I^2}{d\eta^2}\log(2d/\xi) + 1)$ *queries to a* $(\sigma_I, \delta)$-ISGO $\mathcal{O}_I(\cdot)$, *with success probability at least* $1 - \xi - \delta dK$ *and without access to the input* $u$.

With $\eta \leftarrow \epsilon/(4R)$, Lemma 1 implies that $\tilde{O}(\frac{R^2\sigma_I^2}{d\epsilon^2} + 1)$ queries are enough to implement an $(\epsilon/(4R), \epsilon\sqrt{d}/(4R))$-MAGO, which suffices to ensure $x_t$ is $3\epsilon$-optimal by the above analysis while also maintaining a polynomial bound on $\Gamma$. Critically, note that implementing each MAGO query does not require knowledge of the second argument $u$, which is important since the second argument to the MAGO depends on $x^\star$ in the above analysis. We also emphasize that the term ⑤ above cannot

be bounded by attempting to control the error of the MAGO in the direction $x^\star - z$; this is because $z$ is not fixed in advance and indeed *depends* on $g_t$.

Thus, leaving details regarding handling $\delta$ to Section A, combining Lemma 1 with an $\tilde{O}(d)$-algorithm for the feasibility problem suffices to ensure an iterate $x_t$ is $3\epsilon$-optimal using $\tilde{O}(R^2\sigma_I^2/\epsilon^2 + d)$-queries to an ISGO. However, it is not a priori clear which of the $\tilde{O}(d)$ iterates returned by this algorithm is $3\epsilon$-optimal. [53] solve an analogous problem in their setting via a post-processing procedure which iteratively refines the output of the first stage via binary search to ultimately return an $O(\epsilon)$-optimal point (see also [33, 18, 5] for related procedures). In Section A, we carefully adapt this procedure to a $(\sigma_I, \delta)$-ISGO, showing it can also be implemented with $\tilde{O}(R^2\sigma_I^2/\epsilon^2 + d)$ queries and thereby achieving a $d$-factor savings over using a VSGO as in the first stage described above. Finally, at the end of Section A we instantiate Theorem 1 to achieve a new state-of-the-art $\tilde{O}(R^2\sigma_E^2/\epsilon^2 + d)$-complexity for Problem 1 given a $\sigma_E$-ESGO (see Corollary 3), and also recover the aforementioned $\tilde{O}(dR^2\sigma_V^2/\epsilon^2 + d)$ rate given a $\sigma_V$-VSGO as a simple consequence (see Corollary 4).

## 2.2 Overview of our lower bounds for SCO

The starting point of our lower bounds for SCO with ISGOs and ESGOs is the connection between SCO and multivariate mean estimation problems, which has been widely used to derive lower bounds for SCO in various settings (see e.g., [23, 53]). Specifically, as detailed in Section B.2, for any random variable $X$, we consider an instance of Problem 1 where $f$ is defined as the expectation of a collection of linear functions with an added regularizer term. The set of linear functions is parameterized by samples from $X$, which also determines the stochastic gradient, ensuring that it follows a noise model of the same form as $X$. Moreover, finding an $\epsilon$-optimal point of $f$ yields a $\tilde{O}(\epsilon/R)$-estimate of $\mathbb{E}[X]$, thus establishing a reduction from the mean estimation problem to SCO.

Building on this reduction, we establish a lower bound for SCO with ESGOs by introducing a variant of the mean estimation problem where the noise follows a sub-exponential distribution. Specifically, we construct a hard instance in which the random variable $X$ is parameterized by another random variable $V$, as detailed in Section B.1. We show that any algorithm approximating $\mathbb{E}[X]$ must retain significant mutual information with $V$. We then upper bound this mutual information in terms of the number of samples, yielding the desired lower bound for the mean estimation problem (see Lemma 11) and, consequently, for SCO with ESGOs (see Theorem 5). As a corollary, we derive a lower bound for SCO with ISGOs (see Corollary 2). Regarding the mean estimation problem itself (namely, Problem 3), we note that similar lower bounds to Lemma 11 may be derived as corollaries of recent high-probability minimax lower bounds [45], but we provide our own analysis for completeness.

## 2.3 Quantum isotropifier and quantum SCO under variance-bounded noise

Our main technical ingredient of our improved bound for quantum SCO is an unbiased quantum multivariate mean estimation algorithm whose error is isotropic in the sense that it is small in every direction with high probability, which we refer to as *quantum isotropifier*. Adopting the notation of [53, Definition 1], we define having *quantum access to a $d$-dimensional random variable $X$* as the ability to query a *quantum sampling oracle* that produces a quantum superposition representing the probability distribution of $X$.

**Definition 10** (Quantum sampling oracle). *For a $d$-dimensional random variable $X$, its quantum sampling oracle $\mathcal{O}_X$ is defined as*

$$\mathcal{O}_X \ket{0} \to \int_{x \in \mathbb{R}^d} \sqrt{p_X(x)\mathrm{d}x} \ket{x} \otimes \ket{\mathrm{garbage}(x)}, \tag{5}$$

*where $p_X(\cdot)$ represents the probability density function of $X$.*

Using $\tilde{O}(\sigma/\epsilon)$ queries to a quantum sampling oracle $\mathcal{O}_X$ of any random variable $X \in \mathbb{R}^d$ with variance $\sigma^2$, our quantum isotropifier outputs an unbiased estimate $\hat{\mu}$ of $\mathbb{E}[X]$ such that for any unit vector $v \in \mathbb{R}^d$, $|\langle v, \hat{\mu} - \mathbb{E}[X]\rangle|$ is at most $\epsilon$ with high probability. This allows us to construct a $(\sigma_I, \delta)$-ISGO using $\tilde{O}(\sigma_V \mathrm{poly}\log(1/\delta/\sigma_I))$ queries to a $\sigma_V$-QVSGO. Combined with our stochastic cutting plane result in Theorem 1, and with appropriate choices of $\sigma_I$ and $\delta$, this leads to our

improved $\tilde{O}(dR\sigma_V/\epsilon)$ query complexity for quantum SCO in Theorem 7, whose proof can be found in Section C.5.

Next, we provide a technical overview of our quantum isotropifier, which builds on the quantum multivariate mean estimation framework of [22]. We begin by considering a simplified scenario where the random variable $X$ is bounded and satisfies $\|X\| \le 1$. In this case, [22] first implements a directional mean oracle that approximately maps $|g\rangle$ to $e^{i\langle g,\mathbb{E}[X]\rangle}|g\rangle$ for a query $g \in \mathbb{R}^d$. This oracle requires only a constant number of queries to the quantum sampling oracle $\mathcal{O}_X$ provided that $|\mathbb{E}\langle g, X\rangle| \le \|\mathbb{E}[X]\|$. To estimate $\mathbb{E}[X]$ using this directional mean oracle, [22] applies it to the superposition $\sum_{g \in G_m^{\otimes d}}|g\rangle$, where $G_m^{\otimes d}$ is a $d$-dimensional grid defined as

$$G_m = \left\{ \frac{k}{m} - \frac{1}{2} + \frac{1}{2m} : k \in \{0, \ldots, m-1\} \right\} \subset \left( -\frac{1}{2}, \frac{1}{2} \right).$$

The resulting quantum state approximately decomposes as a tensor product

$$\bigotimes_{j=1,\ldots,d} \left( \sum_{g_j \in G_m} e^{ig_j \mathbb{E}[X_j]}|g_j\rangle \right).$$

Then, phase estimation can be performed independently in each dimension to estimate each coordinate of $\mathbb{E}[X]$, similar to the quantum gradient estimation algorithm in [36].

The error in this procedure has two parts: the error in the quantum directional mean oracle and the error from quantum phase estimation. The first part of the error arises as the directional mean oracle is accurate only for grid points $g \in G_m^d$ with $|\langle g, \mathbb{E}[X]\rangle| \le \|\mathbb{E}[X]\|$. In Section C.2, we provide an improved error analysis, showing that only an exponentially small fraction of $g \in G_m^{\otimes d}$ do not satisfy this condition. Hence, in the analysis we can effectively replace the directional mean oracle by a perfect one that performs the map $|g\rangle \to e^{i\langle g,\mathbb{E}[X_j]\rangle}|g\rangle$ exactly for all $g \in G_m^d$, and the resulting quantum state only has an exponentially small change in trace distance.

As for the second part of the error, we note that while the phase estimation procedures in different coordinates are independent, i.e., for any $j \ne k$, the estimates $\hat{\mu}_j$ and $\hat{\mu}_k$ of $\mathbb{E}[X_j]$ and $\mathbb{E}[X_k]$ are sampled independently from two distributions, their biases may still contribute positively in certain directions. Consequently, even if each $\hat{\mu}_j$ has a bounded error satisfying $|\hat{\mu}_j - \mathbb{E}[X_j]| \le \epsilon$, there may exist a unit vector $u \in \mathbb{R}^d$ such that $|\langle \hat{\mu} - \mathbb{E}[X], u\rangle| = \Theta(\epsilon\sqrt{d})$ which is larger than the per-coordinate guarantee by a factor of $\sqrt{d}$. In this work, we demonstrate that the additional $\sqrt{d}$ overhead can be avoided by replacing the original phase estimation with the boosted unbiased phase estimation algorithm from [57]. For each coordinate $j$, this algorithm produces an unbiased estimate $\hat{\mu}_j$ that has bounded error $|\hat{\mu}_j - \mathbb{E}[X_j]| \le \epsilon$ with high probability. Consequently, for any unit vector $u \in \mathbb{R}^d$, the error $\langle \hat{\mu} - \mathbb{E}[X], u\rangle$ can be approximated as a weighted sum of zero-mean bounded random variables, which follows a sub-Gaussian distribution with variance $\Theta(\epsilon^2)$ [61]. This implies that $\hat{\mu}$ not only is unbiased but also has isotropic noise.

We then extend this result to unbounded random variables, as detailed in QUnbounded (Algorithm 3) in Section C.3. Following a similar approach to [22], we decompose $X$ into a sequence of truncated bounded random variables and estimate each one independently. We show that the isotropic noise property is maintained throughout this process. Despite our use of the unbiased phase estimation subroutine [57], the output of QUnbounded may still be biased due to truncations. Notably, while [53] developed a general debiasing scheme for quantum mean estimation algorithms that treats them as black-box procedures, directly applying their scheme to QUnbounded would compromise the isotropic noise property. As detailed in Section C.4, we utilize the multi-level Monte Carlo (MLMC) technique [26] to address this issue. Specifically, we develop a modified version of the MLMC variants introduced in [10, 4, 53] that preserves isotropic noise while maintaining unbiasedness.

To illustrate our approach, suppose we want an unbiased estimate whose error is smaller than $\epsilon$ in any direction with high probability. Let $\hat{\mu}^{(j)}$ denote the estimate obtained by running QUnbounded to accuracy $\beta_j\epsilon$, i.e., $|\langle \hat{\mu}^{(j)} - \mathbb{E}[X], u\rangle| \le \beta_j\epsilon$ with high probability, where $\beta_j$ is a chosen coefficient. Then, our new estimator $\hat{\mu}$ is:

$$\text{Draw } j \sim \text{Geom}\left(\frac{1}{2}\right) \in \mathbb{N}, \text{ compute } \hat{\mu} \leftarrow \hat{\mu}^{(0)} + 2^j(\hat{\mu}^{(j)} - \hat{\mu}^{(j-1)}).$$

This estimator is unbiased as long as $\lim_{j\to\infty} \beta_j = 0$, given that

$$\mathbb{E}[\hat{\mu}] = \mathbb{E}[\hat{\mu}^{(0)}] + \sum_{j=1}^{\infty} \mathbb{P}\{J = j\} 2^j (\mathbb{E}[\hat{\mu}^{(j)}] - \mathbb{E}[\hat{\mu}^{(j-1)}]) = \lim_{j\to\infty} \mathbb{E}[\hat{\mu}^{(j)}] = \mathbb{E}[X].$$

Moreover, the expected query complexity of the new estimator $\hat{\mu}$ is larger than that of $\hat{\mu}^{(0)}$, which is of order $\tilde{O}(\sigma/\epsilon)$, by a multiplicative factor of

$$1 + \sum_{j=1}^{\infty} \mathbb{P}\{J = j\}(\beta_j^{-1} + \beta_{j-1}^{-1}) = 1 + \sum_{j=1}^{\infty} 2^{-j}(\beta_j^{-1} + \beta_{j-1}^{-1}).$$

This factor is a constant if we choose $\beta_j = 2^{-j+C\cdot\log j}$ for any $C > 1$. Furthermore, we show later that $\hat{\mu}$ also satisfies $|\langle \hat{\mu} - \mathbb{E}[X], u\rangle| \leq O(\epsilon)$ with high probability as long as $C$ is a constant, preserving the isotropic noise property. In our full algorithm, Algorithm 4 in Section C.4, we choose $C = 2$ for simplicity.

# 3  Conclusion

We define a new gradient noise model for SCO, termed *isotropic noise,* for which we achieve tight upper and lower bounds (up to polylogarithmic factors). Our upper bound improves upon the state-of-the-art (and in particular SGD) in certain regimes, and as a corollary we achieve a new state-of-the-art complexity for sub-exponential noise. We then develop a subroutine which may be of independent interest called a *quantum isotropifier*, which converts a variance-bounded quantum sampling oracle into an unbiased estimator with isotropic noise. By combining our results, we obtain improved dimension-dependent rates for quantum SCO.

One limitation of our work is we only resolve Problem 1 under stronger assumptions on the noise (e.g., sub-exponential noise in Corollary 3), whereas the rate we achieve under a VSGO (Corollary 4) is worse by roughly a $d$-factor. Another limitation is that our improved quantum rates are dimension-dependent, and have unclear long-term practical impact. Regarding the former, we note that a series of works have shown that dimensions-dependence is necessary for quantum algorithms to achieve an improved scaling in $1/\epsilon$ in a variety of settings [24, 25, 66].

We hope that by identifying the natural open problem Problem 1 and developing techniques to resolve it under stronger assumptions, we leave the door open to future work on resolving fundamental trade-offs between the variance, Lipschitz constant, and dimension in SCO. We believe charting these trade-offs is important since it would yield a sharp understanding of the precisions at which algorithms such as SGD are superseded by other methods. Such questions have long been understood in the deterministic setting, and we believe our work is an important step in resolving them under stochasticity and in quantum settings.

## Acknowledgments

Thank you to anonymous reviewers for their feedback. Aaron Sidford was funded in part by a Microsoft Research Faculty Fellowship, NSF CAREER Award CCF-1844855, NSF Grant CCF1955039, and a PayPal research award. Chenyi Zhang was supported in part by the Shoucheng Zhang graduate fellowship.

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

# A  Upper bounds for SCO under ISGO, ESGO, and VSGO oracles

In this section, we prove an upper bound for Problem 1 given a $(\sigma_I, \delta)$-ISGO in Theorem 1, and then instantiate the result to achieve upper bounds for Problem 1 given either a $\sigma_E$-ESGO or $\sigma_V$-VSGO in Corollaries 3 and 4 respectively. Given an ISGO, our algorithm for Problem 1 proceeds in two stages. In Section A.1, we give a stochastic cutting plane method which utilizes a $(\sigma_I, \delta)$-ISGO to return a finite set of points with the guarantee that at least one of the points is $\epsilon$-optimal. Then in Section A.2, we show how to use a $(\sigma_I, \delta)$-ISGO to obtain an approximate minimum among this finite set of points. We note that all results and discussion in Sections A.1 and A.2 are stated in the context of solving Problem 1 given a $(\sigma_I, \delta)$-ISGO, unless we explicitly specify otherwise. Finally, we put everything together and prove Theorem 1 in Section A.3.

## A.1  Finding candidate solutions via a stochastic cutting plane method

In this subsection, we show how to obtain a finite set of points with the guarantee that at least one of them is $\epsilon$-optimal via a stochastic cutting plane method in the setting of Problem 1 given a $(\sigma_I, \delta)$-ISGO. Cutting plane methods solve feasibility problems, for which we will use the formulation of [52, Definition 8.1.2], except we translate from distance bounds in the $\ell_\infty$-norm to distance bounds in the $\ell_2$-norm.

**Definition 7** (Feasibility Problem). *For $R' \geq r > 0$, we define the $(R', r)$-feasibility problem as follows: We are given query access to a (potentially randomized) halfspace oracle which when queried at $x \in B_2(R')$, outputs a vector $g_x \in \mathbb{R}^d \setminus \{0\}$. The goal is to query the oracle at a sequence of points $x_1, \ldots, x_T \in B_2(R')$ such that $B_2(R') \bigcap_{t \in [T]} H_\geq(g_{x_t}, g_{x_t}^\top x_t)$ does not contain a ball of radius $r$ (namely, any set of the form $B_2(r, z)$ for $z \in \mathbb{R}^d$). We call an algorithm a $\mathcal{T}$-algorithm for the $(R', r)$-feasibility problem if it achieves this goal with at most $\mathcal{T}$ queries.*

We give a standard query complexity for the $(R', r)$-feasibility problem in the following proposition. For completeness, we also give a standard short proof using the center of gravity cutting plane method, but we emphasize that our results are agnostic to the choice of cutting plane method used to prove Proposition 1. See, e.g., [56, 40, 34] for other cutting plane methods with similar query complexities (and which may have improved runtimes).

**Proposition 1.** *There exists an $O(d \log(R'/r))$-algorithm for the $(R', r)$-feasibility problem.*

*Proof.* We apply the so-called center of gravity method, e.g., [8]. For iterations $t = 1, 2, \ldots,$ the center of gravity method queries the halfspace oracle at the origin for $t = 1$ (the center of gravity of $B_2(R')$), and at the center of gravity of the set $B_2(R') \bigcap_{k \in [t-1]} H_\geq(g_{x_k}, g_{x_k}^\top x_k)$ for $t > 1$, where $x_k \in \mathbb{R}^d$ denotes the iterate at step $k$ and $g_{x_k} \in \mathbb{R}^d$ denotes the halfspace queried at $x_k$ during step $k$. Recall that the volume of an $d$-dimensional Euclidean ball (equivalently, $\ell_2$-ball) of radius $\beta > 0$ is $h(d) \cdot \beta^d$, where $h$ is some function of the dimension $d$ which won't matter in the following. Then letting $\mathrm{vol}(\cdot)$ denote the volume of a set, we have $\mathrm{vol}(B_2(R')) = h(d) \cdot (R')^d$, and the volume of any Euclidean ball of radius $r$ is $h(d) \cdot r^d$. Letting $S_t := B_2(R') \bigcap_{k \in [t]} H_\geq(g_{x_k}, g_{x_k}^\top x_k)$ for $t \in \mathbb{N}$, Grünbaum's theorem [29] yields $\mathrm{vol}(S_t) \leq (1 - 1/e)^t \cdot \mathrm{vol}(B_2(R'))$, and the result follows.  $\square$

It is well known that solving the feasibility problem where the halfspace oracle is given by the negative gradient of $f$ suffices to obtain a finite set of points such that at least one is $\epsilon$-optimal [47, 40]. [53] demonstrated that this is still possible with stochastic estimates of the gradient, provided that $-\nabla f$ is approximated to sufficiently high accuracy in the $\ell_2$-norm with high probability at each iteration by averaging over multiple samples. Our key observation is that the analysis of [53] mainly relies on controlling the error of the gradient estimator in a single direction, namely the one-dimensional subspace spanned by $x_t - x^\star$, where $x_t$ is the current iterate. A $(\sigma, \delta)$-ISGO with $\delta$ sufficiently small allows us to control this error with fewer samples than, for example, a $\sigma$-VSGO (Definition 3), allowing for savings of up to a multiplicative $d$ factor in the complexity of estimating the gradient at each step when compared to a $\sigma$-VSGO or $\sigma$-BSGO.

Toward formalizing this discussion, we first define a marginal version of the $\eta$-approximate gradient oracle specified in [53, Definition 5]. Note that this definition still allows us to control the error of the estimator in $\ell_2$-norm. This will be necessary to obtain our guarantee, but we will only need to weakly control this error which can be polynomial in $d$.

**Definition 9** (MAGO). $\tilde{g}(\cdot, \cdot)$ *is a* marginal $(\eta \geq 0, \Gamma \geq 0)$-approximate gradient oracle *(*$(\eta, \Gamma)$-MAGO) if when queried at* $x, u \in \mathbb{R}^d$, *the (potentially random) output* $\tilde{g}(x, u) \in \mathbb{R}^d$ *satisfies*

$$\|\tilde{g}(x,u) - \nabla f(x)\| \leq \Gamma \quad and \quad |\langle \tilde{g}(x,u) - \nabla f(x), \tilde{u}\rangle| \leq \eta \ for \ \tilde{u} := u/\|u\|. \tag{4}$$

Next, we show that an $(\eta, \eta\sqrt{d})$-MAGO can be implemented with high probability using a $(\sigma_I, \delta)$-ISGO without knowledge of the second argument, the directional input $u$.

**Lemma 1.** *For any* $\delta, \xi \in (0,1)$; $x, u \in \mathbb{R}^d$; *and* $\eta > 0$, *a query* $\tilde{g}(x,u)$ *to an* $(\eta, \Gamma := \eta\sqrt{d})$-MAGO *can be implemented using* $K = O(\frac{\sigma_I^2}{d\eta^2} \log(2d/\xi) + 1)$ *queries to a* $(\sigma_I, \delta)$-ISGO $\mathcal{O}_I(\cdot)$, *with success probability at least* $1 - \xi - \delta d K$ *and without access to the input* $u$.

*Proof.* For $K \in \mathbb{N}$ to be chosen later, consider the estimator $\overline{Z} := \frac{1}{K} \sum_{k \in [K]} Z_k$ for $Z_1, \ldots, Z_K \overset{\text{iid}}{\sim} \mathcal{O}_I(x)$, which does not require knowledge of $u$. Let $\{u_\ell\}_{\ell \in [d]}$ denote any orthonormal basis of $\mathbb{R}^d$ such that $u_1 = u/\|u\|$, and define for all $\ell \in [d]$:

$$\overline{Y}^{(\ell)} := \frac{1}{K} \sum_{k \in [K]} Y_k^{(\ell)}, \quad \text{where} \ Y_k^{(\ell)} := \langle Z_k - \nabla f(x), u_\ell \rangle \ \text{for} \ k \in [K].$$

Note that $\mathbb{E}Y_k^{(\ell)} = 0$, and $|Y_k^{(\ell)}| \leq \sigma_I/\sqrt{d}$ for all $(k, \ell) \in [K] \times [d]$ with probability at least $1 - \delta d K$ by Definition 4 and a union bound. Then conditioning on the latter event $\mathcal{E}$, Hoeffding's inequality [63, Proposition 2.5] yields for all $\ell \in [d]$:

$$\mathbb{P}\Big[|\overline{Y}^{(\ell)}| \geq t \mid \mathcal{E}\Big] \leq 2\exp\left(\frac{-Kdt^2}{2\sigma_I^2}\right) \quad \text{for all } t \geq 0.$$

Choosing $K = O\Big(\frac{\sigma_I^2}{d\eta^2} \log(2d/\xi) + 1\Big)$ gives $\mathbb{P}\Big[|\overline{Y}^{(\ell)}| \geq \eta \mid \mathcal{E}\Big] \leq \xi/d$ for all $\ell \in [d]$, in which case

$$\mathbb{P}\underbrace{\left[\sup_{\ell \in [d]} |\langle \overline{Z} - \nabla f(x), u_\ell \rangle| \leq \eta\right]}_{=:\mathcal{E}'} = \mathbb{P}\left[\sup_{\ell \in [d]} |\overline{Y}^{(\ell)}| \leq \eta \mid \mathcal{E}\right] \cdot \mathbb{P}[\mathcal{E}] \geq (1 - \xi)(1 - \delta d K) \geq 1 - \xi - \delta d K$$

by a union bound. Finally, note that the event $\mathcal{E}'$ implies the first condition in (4) since then

$$\|\overline{Z} - \nabla f(x)\| = \sqrt{\sum_{\ell \in [d]} \langle \overline{Z} - \nabla f(x), u_\ell \rangle^2} \leq \eta\sqrt{d},$$

and the second condition in (4) follows because recall we chose $u_1 = u/\|u\|$. $\qquad\square$

Next, we show how to obtain a finite set of points such that at least one is $\epsilon$-optimal using an appropriate MAGO. Critically, the second argument to the MAGO must be allowed to depend on $x^\star$. However, this does not pose an issue when we ultimately instantiate the MAGO queries via ISGO queries since doing so does not require any knowledge of the second argument to the MAGO, per Lemma 1.

**Lemma 2** (Obtaining candidate solutions via the feasibility problem). *Suppose* $R', r > 0$ *are such that* $B_2(r, x^\star) \subseteq B_2(R')$ *and* $f(z) \leq f(x^\star) + \epsilon/2$ *for all* $z \in B_2(r, x^\star)$. *Then given an* $(\eta := \frac{\epsilon}{8R'}, \Gamma := \eta\sqrt{d})$-MAGO $\tilde{g}(\cdot, \cdot)$ *where only the second argument can depend on* $x^\star$ *and a* $\mathcal{T}$-algorithm for the $(R', r)$-feasibility problem, and additionally supposing $r \leq \epsilon/(4\Gamma)$, we can compute a finite set of points $S \subseteq B_2(R')$ with $|S| = \mathcal{T}$ such that $\min_{x \in S} f(x) \leq f(x^\star) + \epsilon$.

*Proof.* We run the $\mathcal{T}$-algorithm for the $(R', r)$-feasibility problem with the halfspace oracle at a point $x \in \mathbb{R}^d$ given by $-\tilde{g}(x, x - x^\star)$. By definition, this produces a sequence of points $S := \{x_t \in B_2(R')\}_{t \in [\mathcal{T}]}$ such that $B_2(R') \bigcap_{t \in [\mathcal{T}]} H_{\geq}(-g_t, -g_t^\top x_t)$ does not contain a ball of radius $r$, where $-g_t$ denotes the output of the halfspace oracle at the $t$-th step (i.e., $g_t$ was the result of the oracle call $\tilde{g}(x_t, x_t - x^\star)$). (If the $\mathcal{T}$-algorithm terminates with less than $\mathcal{T}$ queries, we can pad $S$ with

duplicates.) Note that we can terminate immediately if the halfspace oracle outputs $g_t = 0$ at some step $t$ since then

$$f(x_t) - f(x^\star) \le \langle \nabla f(x_t), x_t - x^\star \rangle = \langle \nabla f(x_t) - g_t, x_t - x^\star \rangle \le 2R'\left\langle \nabla f(x_t) - g_t, \frac{x_t - x^\star}{\|x_t - x^\star\|} \right\rangle \le \epsilon$$

by convexity, the fact that $x_t, x^\star \in B_2(R')$, and the definition of $\tilde{g}(x_t, x_t - x^\star)$. Then supposing $g_t \ne 0$ for all $t \in [\mathcal{T}]$, it must be the case that at some iteration $t \in [\mathcal{T}]$, there exists $z \in B_2(r, x^\star)$ such that $-g_t^\top z < -g_t^\top x_t$. Then by convexity

$$\begin{aligned}
f(x_t) &\le f(z) + \langle \nabla f(x_t), x_t - z \rangle \\
&= f(z) + \underbrace{\langle g_t, x_t - z \rangle}_{\text{①}} + \underbrace{\langle \nabla f(x_t) - g_t, x_t - x^\star \rangle}_{\text{②}} + \underbrace{\langle \nabla f(x_t) - g_t, x^\star - z \rangle}_{\text{③}} \le f(x^\star) + \epsilon,
\end{aligned}$$

where the last inequality followed because ① $< 0$ by the definition of $z$; we have ② $\le \eta \|x_t - x^\star\| \le 2R'\eta \le \epsilon/4$ by the definition of $\tilde{g}(x_t, x_t - x^\star)$; we have ③ $\le \Gamma r \le \epsilon/4$ by Cauchy-Schwarz, the definition of $\tilde{g}(x_t, x_t - x^\star)$, and the additional assumption on $r$; and finally $z$ is $\epsilon/2$-optimal. □

Finally, we give our main result for this subsection:

**Lemma 3** (Obtaining candidate solutions via an ISGO). *For $T = O(d\log(d + RL/\epsilon))$, there exists an algorithm which returns a finite set of points $S \subseteq B_2(2R)$ with $|S| = T$ and $\min_{x \in S} f(x) \le f(x^\star) + \epsilon$ using*

$$M = O\left( T\left( \frac{R^2 \sigma_I^2}{d\epsilon^2} \log(2dT/\xi) + 1 \right) \right) \quad \text{queries to a } (\sigma_I, \delta)\text{-ISGO } \mathcal{O}_I(\cdot)$$

*and with success probability at least $1 - \xi - \delta dM$.*

*Proof.* We first apply Lemma 2 with $R' \coloneqq 2R$ and $r \coloneqq \min\{\frac{\epsilon}{2L}, \frac{\epsilon}{4\Gamma}\}$ (with $\eta \coloneqq \frac{\epsilon}{8R'}$ and $\Gamma \coloneqq \eta\sqrt{d}$ as in Lemma 2). Note that we can assume $\epsilon \in (0, RL)$ without loss of generality as otherwise the origin is $\epsilon$-optimal, in which case $R \ge r$, implying $B_2(r, x^\star) \subseteq B_2(R')$. Furthermore, every point in $B_2(r, x^\star)$ is $\epsilon/2$-optimal given that $f$ is $L$-Lipschitz. Thus, combining Lemma 2 with Proposition 1, we can obtain $S$ with

$$T = O(d\log(R'/r)) = O(d\log(R(L+\Gamma)/\epsilon)) = O(d\log(d + RL/\epsilon))$$

queries to a $(\eta = \frac{\epsilon}{8R'}, \Gamma = \eta\sqrt{d})$-MAGO. The result then follows by instantiating each MAGO query per Lemma 1, and the final success probability follows from a union bound. □

## A.2 Approximate minimum finding among a finite set of points

Here we show how to use an ISGO to obtain an approximate minimum among a finite set of points $S \subseteq \mathbb{R}^d$. (As a reminder, Section A.2 is stated in the context of Problem 1 given access to a $(\sigma_I, \delta)$-ISGO, unless we specify otherwise.) To do so, we adapt the techniques of [53, Section 4.2], which solves the same problem except with a (quantum) BSGO (recall Definition 2).[8] At a high level, the procedure involves iteratively replacing pairs of points $(x, x')$ in $S$ with a point $\bar{x}$ on the line segment between $x$ and $x'$, with the guarantee that the objective value of $\bar{x}$ is (approximately) at least as good as that of both $x$ and $x'$. This is done in multiple levels, eventually comparing pairs that were the result of previous comparisons, until only a single point remains. For a pair $(x, x')$, the point $\bar{x}$ is computed via a binary search procedure on the line segment between them using the ISGO. Critically, analogously to the main result of Section A.1, a $(\sigma, \delta)$-ISGO allows for up to a $d$-factor savings when implementing this binary search over a $\sigma$-BSGO or $\sigma$-VSGO.

To start, Algorithm 1, which is stated independently of the context of Problem 1, gives a procedure for computing an approximate minimum over $[0, 1]$ of a one-dimensional Lipschitz convex function, given access to a sequence of approximate first-order derivatives (see Line 4). We will ultimately use Algorithm 1 to obtain $\bar{x}$ for a pair $(x, x')$ as discussed above. Algorithm 1 and its analysis are based on Algorithm 4 in [53].

---

[8]An alternate procedure for solving this problem via a BSGO was given in [33, Proposition 3]. However, unlike the procedure of [53, Section 4.2], it is not clear how to adapt their techniques to an ISGO (or even a VSGO). See also [18, 5] for related procedures.

**Algorithm 1:** InexactLineSearch$(h, \epsilon')$

---

**Input:** Convex $G$-Lipschitz $h : \mathbb{R} \to \mathbb{R}$, target accuracy $\epsilon' > 0$
**Output:** $\bar{z} \in [0, 1]$ s.t. $h(\bar{z}) \leq \min_{z \in [0,1]} h(z) + \epsilon'$

1 $z_\ell \leftarrow 0$, $z_r \leftarrow 1$
2 **while** $z_r - z_\ell > \epsilon'/G$ **do**
3     $z_m \leftarrow (z_r + z_\ell)/2$
4     Let $\tilde{u}_{z_m}$ be s.t. $|\tilde{u}_{z_m} - h'(z_m)| \leq \epsilon'/4$
5     **if** $|\tilde{u}_{z_m}| \leq \epsilon'/4$ **then return** $\bar{z} \leftarrow z_m$
6     **if** $\tilde{u}_{z_m} > 0$ **then** $z_r \leftarrow z_m$ **else** $z_\ell \leftarrow z_m$

7 **return** $\bar{z} \leftarrow z_\ell$

---

**Lemma 4** (Algorithm 1 guarantee). *Given differentiable, convex, $G$-Lipschitz $h \colon \mathbb{R} \to \mathbb{R}$ and $\epsilon' > 0$, Algorithm 1 terminates after $O(\log(G/\epsilon'))$ iterations[9] and returns $\bar{z} \in [0, 1]$ such that $h(\bar{z}) \leq \min_{z \in [0,1]} h(z) + \epsilon'$.*

*Proof.* The iteration bound is immediate from the fact that each iteration halves the length of $[z_\ell, z_r]$. As for correctness, if Algorithm 1 terminates on Line 6, then $|h'(z_m)| \leq \epsilon'/2$ by a triangle inequality, in which case convexity implies for all $z \in [0, 1]$:

$$h(z_m) \leq h(z) + h'(z_m)(z_m - z) \leq h(z) + |h'(z_m)| \, |z_m - z| \leq h(z) + \epsilon'.$$

On the other hand, suppose Algorithm 1 terminates on Line 7. Pick any $z^\star \in \operatorname{argmin}_{z \in [0,1]} h(z)$, and we claim Algorithm 1 maintains the invariant $z^\star \in [z_\ell, z_r]$. This follows by induction because the condition $\tilde{u}_{z_m} > 0$ in Line 6 implies $\tilde{u}_{z_m} > \epsilon'/4$ due to the failure of the termination condition in Line 5, in which case $h'(z_m) > 0$ by a reverse triangle inequality. Then

$$h'(z_m)(z_m - z^\star) \geq h(z_m) - h(z^\star) \geq 0 \implies z_m - z^\star \geq 0 \implies z^\star \in [z_\ell, z_m].$$

The case where $\tilde{u}_{z_m} < 0$ in Line 6 is analogous. To conclude, the termination condition of Line 2 implies the following in Line 7 by the invariant and Lipschitzness:

$$z_r - z_\ell \leq \epsilon'/G \implies |z_\ell - z^\star| \leq \epsilon'/G \implies h(z_\ell) \leq h(z^\star) + \epsilon'.$$

$\square$

Before proceeding, we provide a version of Lemma 1 where we don't control the error of the MAGO estimator in $\ell_2$-norm, allowing for a potential logarithmic-factor improvement in the ISGO query complexity needed to implement the MAGO as well as a higher success probability. Technically Lemma 5 is not necessary to prove our ultimate guarantee Theorem 1, where we do not explicitly state polylogarithmic factors for brevity, but we include it for completeness and to make it clear what aspects of the MAGO we actually need for individual lemmas.

**Lemma 5.** *For any $\delta, \xi \in (0, 1)$; $x, u \in \mathbb{R}^d$; and $\eta > 0$, a query $\tilde{g}(x, u)$ to an $(\eta, \infty)$-MAGO can be implemented using $K = O\left(\frac{\sigma_I^2}{d\eta^2} \log(2/\xi) + 1\right)$ queries to a $(\sigma_I, \delta)$-ISGO $\mathcal{O}_I(\cdot)$, with success probability at least $1 - \xi - \delta K$ and without access to the input $u$.*

*Proof.* Consider the estimator $\overline{Z} := \frac{1}{K} \sum_{k \in [K]} Z_k$ for $Z_1, \ldots, Z_K \overset{\text{iid}}{\sim} \mathcal{O}_I(x)$, which doesn't require knowledge of $u$. Define

$$\overline{Y} := \frac{1}{K} \sum_{k \in [K]} Y_k \quad \text{where} \quad Y_k := \langle Z_k - \nabla f(x), u \rangle \text{ for } k \in [K].$$

Note $\mathbb{E} Y_k = 0$ and $|Y_k| \leq \sigma_I/\sqrt{d}$ for all $k \in [K]$ with probability at least $1 - \delta K$ by Definition 4 and a union bound. Letting $\mathcal{E}$ denote the latter event, an application of Hoeffding's inequality implies $\mathbb{P}\big[|\overline{Y}| \geq \eta \mid \mathcal{E}\big] \leq \xi$, in which case

$$\mathbb{P}\big[\langle \overline{Z} - \nabla f(x), u \rangle \leq \eta\big] = \mathbb{P}\big[|\overline{Y}| \leq \eta \mid \mathcal{E}\big] \cdot \mathbb{P}[\mathcal{E}] \geq 1 - \xi - \delta K.$$

$\square$

---

[9] An iteration is a single execution of Lines 3–6.

We now use Algorithm 1 as a subroutine to obtain $\bar{x}$ for a pair $(x, x')$.

**Lemma 6.** *For $\epsilon' > 0$; $\xi \in (0, 1)$; and $x, x' \in \mathbb{R}^d$ with $D := \|x' - x\|$, there exists an algorithm which computes $\bar{x}$, a convex combination of $x$ and $x'$ such that $f(\bar{x}) \leq \min_{\lambda \in [0,1]} f(x + \lambda(x' - x)) + \epsilon'$ using at most*

$$M = O\left(M'\left(\frac{D^2 \sigma_I^2}{d\epsilon'^2} \log(M'/\xi) + 1\right)\right) \text{ for } M' = O(\log(DL/\epsilon'))$$

*queries to a $(\sigma_I, \delta)$-ISGO $\mathcal{O}_I(\cdot)$, with success probability at least $1 - \xi - \delta M$.*

*Proof.* If $D = 0$, we return $x$. Otherwise, define $x_\lambda := x + \lambda(x' - x)$ for $\lambda \in \mathbb{R}$, and let $h(\lambda) := f(x_\lambda)$. Note that $h$ is $DL$-Lipschitz since

$$h'(\lambda) = \langle \nabla f(x_\lambda), x' - x \rangle \implies |h'(\lambda)| \leq \|\nabla f(x_\lambda)\| \, \|x' - x\| \leq DL.$$

Note that for a given $\lambda \in \mathbb{R}$ and $\gamma > 0$, we can obtain $\tilde{u}_\lambda \in \mathbb{R}$ such that $|\tilde{u}_\lambda - h'(\lambda)| \leq \gamma$ using a single query to a $(\gamma/D, \infty)$-MAGO $\tilde{g}(\cdot, \cdot)$. Indeed, with $Z \sim \tilde{g}(x_\lambda, x' - x)$ and $\tilde{u}_\lambda \leftarrow \langle Z, x' - x \rangle$,

$$|\tilde{u}_\lambda - h'(\lambda)| = |\langle Z - \nabla f(x_\lambda), x' - x \rangle| \leq \gamma.$$

Furthermore, for $\xi' \in (0, 1)$, each query to $\tilde{g}(\cdot, \cdot)$ can be implemented with $K = O\left(\frac{D^2 \sigma_I^2}{d\gamma^2} \log(2/\xi') + 1\right)$ queries to a $(\sigma_I, \delta)$-ISGO and success probability $1 - \xi' - \delta K$ by Lemma 5.

Putting this together, we can obtain $\tilde{u}_\lambda$ such that $|\tilde{u}_\lambda - h'(\lambda)| \leq \gamma$ with $K = O\left(\frac{D^2 \sigma_I^2}{d\gamma^2} \log(2/\xi') + 1\right)$ queries to a $(\sigma_I, \delta)$-ISGO and success probability $1 - \xi' - \delta K$. Combining this with the fact that $h$ is $DL$-Lipschitz, it is clear that $\bar{\lambda} \leftarrow \texttt{InexactLineSearch}(h, \epsilon')$ can be implemented with

$$M = O\left(M'\left(\frac{D^2 \sigma_I^2}{d\epsilon'^2} \log(M'/\xi) + 1\right)\right) \text{ for } M' = O(\log(DL/\epsilon'))$$

queries to a $(\sigma_I, \delta)$-ISGO, with, by a union bound, success probability at least $1 - \xi - \delta M$. To conclude, note

$$f(\bar{x} := x_{\bar{\lambda}}) = h(\bar{\lambda}) \leq \min_{\lambda \in [0,1]} h(\lambda) = \min_{\lambda \in [0,1]} f(x + \lambda(x' - x)).$$

$\square$

Finally, we give our main result for this subsection:

**Lemma 7** (Finding the best among a finite set of points). *For $T \in \mathbb{N}$, suppose $S := \{x_t \in \mathbb{R}^d\}_{t \in [T]}$ is such that $\|x_i - x_j\| \leq D$ for all $i, j \in [T]$. Then for $\epsilon > 0$ and $\xi \in (0, 1)$, there is an algorithm which returns a point $x$ in the convex hull of $S$ such that $f(x) \leq \min_{x' \in S} f(x') + \epsilon$ using at most $M(T - 1)$ queries to a $(\sigma_I, \delta)$-ISGO $\mathcal{O}_I(\cdot)$, for*

$$M = O\left(M'\left(\frac{D^2 \sigma_I^2}{d\epsilon^2} \log^2(T) \log(M' \log T/\xi) + 1\right)\right) \text{ with } M' = O(\log(DL \log T/\epsilon)),$$

*and it succeeds with probability at least $1 - \xi - \delta M \log T$*

*Proof.* For a pair $x, x'$ in the convex hull of $S$, we can apply Lemma 6 to obtain $\bar{x}$, a convex combination of $x$ and $x'$ such that $f(\bar{x}) \leq \min\{f(x), f(x')\} + \epsilon/\log T$ using

$$M = O\left(M'\left(\frac{D^2 \sigma_I^2}{d\epsilon^2} \log^2(T) \log(M' \log T/\xi) + 1\right)\right) \text{ for } M' = O(\log(DL \log T/\epsilon))$$

queries to a $(\sigma_I, \delta)$-ISGO, with success probability $1 - \xi/\log T - \delta M$.

Next, we assume without loss of generality that $|T|$ is a power of 2, as $x_1$ can be duplicated if this is not the case. Then consider a complete binary tree with $T$ leaves, and define the $\ell$-th layer for $\ell \in \{0\} \cup [\log_2 T]$ to denote those vertices which are distance $\ell$ from a leaf node. (In other words, the 0-th layer contains the leaf nodes, and the final $(\log_2 T)$-th layer is the root.) We assign $x_1, \dots, x_T$ to the leaf nodes, and iteratively populate the $\ell$-th layer of the tree for $\ell = 1, 2, \dots, \log_2 T$

via the following process: For each node in the $\ell$-th layer with children $x$ and $x'$ in the $(\ell + 1)$-th layer, we assign to that node the estimator described above with inputs $x$ and $x'$. Assuming $x_1 \in \arg\min_{x \in S} f(x)$ without loss of generality, note that conditioned on all of the estimates along the path from the leaf $x_1$ to the root succeeding, which happens with probability at least $1 - \xi - \delta M \log T$ by a union bound, the value of $f$ at the root $x_r$ (which we output) can be bounded as $f(x_r) \leq f(x_1) + \epsilon$. The final query bound follows from the fact that we call the estimator $T - 1$ times, since there are $T - 1$ non-leaf nodes in the tree. $\qquad \square$

### A.3 Putting it all together

We now restate and prove our main guarantee for Problem 1 given a $(\sigma_I, \delta)$-ISGO:

**Theorem 1.** *Problem 1 can be solved with probability at least $2/3$ in $\tilde{O}\left(R^2 \sigma_I^2 / \epsilon^2 + d\right)$ queries to a $(\sigma_I, \delta)$-ISGO for any $\delta \leq \frac{1}{Md(R^2 \sigma_I^2 / \epsilon^2 + d)}$, where $M = \tilde{O}(1)$.*

*Proof.* By Lemma 3, we can obtain a finite set $S \subseteq B_2(2R)$ with $|S| = \tilde{O}(d)$ and $\min_{x \in S} f(x) \leq f(x^\star) + \epsilon/2$ using $K = \tilde{O}(R^2 \sigma_I^2 / \epsilon^2 + d)$ queries to the $(\sigma_I, \delta)$-ISGO, with success probability at least $9/10 - \delta d K$. Then giving this set $S$ as input to Lemma 7, we can obtain $x \in B_2(2R)$ such that $f(x) \leq \min_{x' \in S} f(x') + \epsilon/2$ using $K' = \tilde{O}\left(R^2 \sigma_I^2 / \epsilon^2 + d\right)$ queries to the $(\sigma_I, \delta)$-ISGO, with success probability at least $9/10 - \delta \cdot \tilde{O}\left(\frac{R^2 \sigma_I^2}{d\epsilon^2} + 1\right)$. Then $f(x) \leq f(x^\star) + \epsilon$ with probability at least $2/3$ by the valid range of $\delta$ and a union bound, and the total number of queries is $K + K' = \tilde{O}(R^2 \sigma_I^2 / \epsilon^2 + d)$. $\qquad \square$

Next, we instantiate Theorem 1 to obtain upper bounds for solving Problem 1 with a $\sigma_E$-ESGO or $\sigma_V$-VSGO. First, we show in the following lemmas that ISGOs can be implemented using ESGOs and VSGOs respectively. In particular, Lemma 8 says that an ISGO query can be implemented with an ESGO at the cost of only a log factor, whereas Lemma 9 says that doing so with a VSGO additionally picks up a $\sqrt{d}$ factor. Indeed, this $\sqrt{d}$-factor is the reason our rate for solving Problem 1 with a VSGO is a $d$-factor worse than solving Problem 1 with an ISGO or ESGO. We note that the proof of Lemma 9 is based on ideas from [53, Lemma 7].

**Lemma 8.** *For any $\sigma_E > 0$ and $\delta \in (0, 1)$, a $\sigma_E$-ESGO is a $(\sigma_E \log(2/\delta), \delta)$-ISGO.*

*Proof.* For any $\delta \in (0, 1)$ and any unit vector $u \in \mathbb{R}^d$, by Eq. 2 we have

$$\mathbb{P}\left[|\langle \mathcal{O}_E(x) - \nabla f(x), u\rangle| \geq (\sigma_E / \sqrt{d}) \cdot \log(2/\delta)\right] \leq \delta.$$

$\square$

**Lemma 9.** *For any $\sigma_V > 0$ and $\delta \in (0, 1)$, a query to a $(6\sigma_V \sqrt{d}, \delta)$-ISGO $\mathcal{O}_I(\cdot)$ can be implemented using $O(\log(2/\delta))$ queries to a $\sigma_V$-VSGO $\mathcal{O}_V(\cdot)$.*

*Proof.* Supposing we wish to query $\mathcal{O}_I(\cdot)$ at $x \in \mathbb{R}^d$, let $Z_1, \ldots, Z_K \overset{\text{iid}}{\sim} \mathcal{O}_V(x)$ for $K \in \mathbb{N}$ to be chosen later. Chebyshev's inequality yields $\mathbb{P}[\|Z_k - \nabla f(x)\| \geq 2\sigma_V] \leq 1/4$ for all $k \in [K]$. Define $J := \{k \in [K] : \|Z_k - \nabla f(x)\| \leq 2\sigma_V\}$, in which case a Chernoff bound gives $\mathbb{P}[|J| \leq 2K/3] \leq 2e^{-cK}$ for some absolute constant $c \in (0, 1)$. Thus, choosing $K = O(\log(2/\delta))$ yields $\mathbb{P}[|J| \leq 2K/3] \leq \delta$. Then conditioning on the event where $|J| \geq 2K/3$, which happens with probability at least $1 - \delta$, observe that if a sample $Z_k$ for some $k \in [K]$ is such that

$$|\{k' \in [K] : \|Z_k - Z_{k'}\| \leq 4\sigma_V\}| \geq 2K/3,$$

namely $Z_k$ is $4\sigma_V$-close to at least $2K/3$ of $Z_1, \ldots, Z_K$, then there must exist some $j \in J$ such that $\|Z_k - Z_j\| \leq 2\sigma_V$, implying $\|Z_k - \nabla f(x)\| \leq 6\sigma_V$ by a triangle inequality. Furthermore, such a sample $Z_k$ is guaranteed to exist when conditioning on $|J| \geq 2K/3$ since any sample corresponding to an index in $J$ satisfies this property in particular by a triangle inequality. Finally, we conclude by noting therefore that when $|J| \geq 2K/3$, we have for any $u \in \mathbb{R}^d$ with $\|u\| = 1$:

$$|\langle Z_k - \nabla f(x), u\rangle| \leq \|Z_k - \nabla f(x)\| \leq 6\sigma_V.$$

$\square$

Finally, we give our upper bounds for Problem 1 with a $\sigma_E$-ESGO or $\sigma_V$-VSGO in the following corollaries, which we restate here for convenience.

**Corollary 3.** *Problem 1 can be solved with probability at least $2/3$ in $\tilde{O}(R^2\sigma_E^2/\epsilon^2 + d)$ queries to a $\sigma_E$-ESGO.*

*Proof.* By Lemma 8, $\mathcal{O}_E(\cdot)$ is a $(\sigma_E \log(2/\delta), \delta)$-ISGO per Definition 4. The result then follows by setting $\delta \leftarrow \frac{1}{Md(R^2\sigma_E^2/\epsilon^2+d)}$ where $M = \tilde{O}(1)$ and applying Theorem 1. $\qquad\square$

**Corollary 4.** *Problem 1 can be solved with probability at least $2/3$ in $\tilde{O}(dR^2\sigma_V^2/\epsilon^2 + d)$ queries to a $\sigma_V$-VSGO.*

*Proof.* By Lemma 9, we can implement a query to a $(6\sigma_V\sqrt{d}, \delta)$-ISGO with $O(\log(1/\delta))$ queries to $\mathcal{O}_V(\cdot)$. We conclude by setting $\delta \leftarrow \frac{1}{Md^2(R^2\sigma_V^2/\epsilon^2+d)}$ where $M = \tilde{O}(1)$ and applying Theorem 1. $\qquad\square$

# B    Lower bounds for SCO under ISGO and ESGO oracles

In this section, we establish lower bounds for solving Problem 1 with either a $(\sigma_I, \delta)$-ISGO or $\sigma_E$-ESGO. We begin by defining a mean estimation problem for random variables with sub-exponential noise and provide a lower bound for this problem in Section B.1. Then in Section B.2, we establish our lower bound for solving Problem 1 given a $\sigma_E$-ESGO by reducing this isotropic mean estimation problem to the former. Finally, we give our lower bound for Problem 1 given a $(\sigma_I, \delta)$-ISGO at the end of Section B.2 as a simple corollary by noting that a $\sigma$-ESGO is a $\tilde{O}(\sigma \log(2/\delta), \delta)$-ISGO for any $\delta \in (0, 1)$.

## B.1    Lower bound for mean estimation with sub-exponential noise

**Problem 3** (Mean estimation with sub-exponential noise). *Given sample access to a $d$-dimensional random variable $X$ that satisfies*

$$\mathbb{P}[|\langle u, X - \mathbb{E}[X]\rangle| \geq t] \leq 2\exp\left(-t\sqrt{d}/\sigma_E\right)$$

*for any $t > 0$ and any unit vector $u$, the goal is to output an estimate $\hat{\mu}$ of $\mu := \mathbb{E}[X]$ satisfying $\|\hat{\mu} - \mu\|_2 \leq \tilde{\epsilon}$.*

Let $v \in \{\pm 1\}^d$ be a fixed vector in the $d$-dimensional hypercube. We establish our lower bound for Problem 3 by considering the following $d$-dimensional random variable $X$, where each component $X_i$ are sampled independently at random

$$X_i = \begin{cases} \frac{v_i\sigma_E}{2\sqrt{d}}, & w.p. \ \frac{1}{2} + \frac{8\tilde{\epsilon}\sqrt{\log d}}{\sigma_E}, \\ -\frac{v_i\sigma_E}{2\sqrt{d}}, & w.p. \ \frac{1}{2} - \frac{8\tilde{\epsilon}\sqrt{\log d}}{\sigma_E}. \end{cases} \qquad (6)$$

for all $i \in [d]$. Let $P_v(\cdot)$ denote this probability distribution.

**Lemma 10.** *For any fixed $v \in \{\pm 1\}^d$ and any unit vector $u \in \mathbb{R}^d$, the random variable $X$ defined in Eq. 6 satisfies*

$$\mathbb{P}[|\langle u, X - \mathbb{E}[X]\rangle| \geq t] \leq \max\left\{2\exp\left(-t^2 d/\sigma_E^2\right), 1\right\} \leq 2\exp\left(-t\sqrt{d}/\sigma_E\right).$$

*Proof.* For any unit vector $u \in \mathbb{R}^d$, using the fact that each coordinate $X_j$ is sampled independently and satisfies $|X_j - \mathbb{E}[X_j]| \leq \sigma_E/\sqrt{d}$, by Lemma 21 we know that $\langle u, X - \mathbb{E}[X]\rangle$ is a sub-Gaussian random variable with variance

$$\sum_{j=1}^{d} \frac{u_i^2\sigma_E^2}{d} = \sigma_E^2/d.$$

Hence, for any $t > 0$ we have

$$\mathbb{P}[|\langle u, X - \mathbb{E}[X]\rangle| \geq t] \leq \max\left\{2\exp\left(-t^2 d/\sigma_E^2\right), 1\right\} \leq 2\exp\left(-t\sqrt{d}/\sigma_E\right).$$

$\qquad\square$

**Lemma 11.** *Any algorithm that solves Problem 3 with success probability at least $2/3$ must have observed at least*

$$\Omega\Big(\frac{\sigma_E^2}{\tilde{\epsilon}^2 \log d}\Big)$$

*i.i.d. samples of $X$.*

**Lemma 12** (Chain rule of mutual information, Theorem 2.5.2 of [55])**.** *For any $n > 0$ and any random variables $V, X^{(1)}, \ldots, X^{(n)}$, we have*

$$I(V; X^{(1)}, \ldots, X^{(n)}) = \sum_{i=1}^{n} I(V; X^{(i)} | X^{(1)}, \ldots, X^{(i-1)}).$$

*Proof of Lemma 11.* Consider the random variable $X \sim P_v$ defined in Eq. 6 where $v$ is chosen uniformly at random from $\{\pm 1\}^d$. For convenience, let $p = 8\tilde{\epsilon}\sqrt{\log d}/\sigma_E$ and $\tilde{\sigma}_E = \frac{\sigma_E}{2\sqrt{d}}$. Observe that $V \to (X^{(1)}, \ldots, X^{(n)}) \to \hat{X}$ is a Markov chain, regardless of the algorithm used to construct $\hat{X}$. Therefore, by the data processing inequality,

$$I(V; \hat{X}) \leq I(V; (X^{(1)}, \ldots, X^{(n)})). \tag{7}$$

where $I(V; \hat{X})$ is the mutual information between $V$ and $\hat{X}$, as defined in Section 1.3. We will prove that if an algorithm outputs $\hat{X}$ such that $\|\hat{X} - \mathbb{E}_{P_v}[X]\| \leq \tilde{\epsilon}$ with success probability at least $2/3$ then,

$$I(V; \hat{X}) \geq d/6. \tag{8}$$

On the other hand, we will prove that

$$I(V; (X^{(1)}, \ldots, X^{(n)})) \leq 28ndp^2. \tag{9}$$

Assuming Eq. 8 and Eq. 9 for now and combining them with Eq. 7 we have $d/6 \leq 28ndp^2$. Therefore,

$$n \geq \frac{1}{6 \cdot 28p^2} = \Omega\Big(\frac{\sigma_E^2}{\tilde{\epsilon}^2 \log d}\Big).$$

Proof of Eq. 8: Suppose an algorithm outputs $\hat{X}$ satisfying $\|\hat{X} - \mathbb{E}_{P_v}[X]\| \leq \tilde{\epsilon}$ with success probability at least $2/3$. Let $E$ be the event that $\|\hat{X} - \mathbb{E}_{P_v}[X]\| \leq \tilde{\epsilon}$. We have

$$
\begin{aligned}
I(V; \hat{X}) &= H(V) - H(V|\hat{X}) \\
&= H(V) - \Big(H(V|\hat{X}, E)p(E) + H(V|\hat{X}, \neg E)(1 - p(E))\Big) \\
&\geq H(V) - \Big(H(V|\hat{X}, E)p(E) + H(V)(1 - p(E))\Big) \\
&= p(E)\Big(H(V) - H(V|\hat{X}, E)\Big).
\end{aligned}
$$

Observe that $p(E) \geq 2/3$ and $H(V) = d$. To compute $H(V|\hat{X}, E)$ we consider the set

$$S := \Big\{v \in \{\pm 1\}^d \text{ s.t. } \Big\|\hat{X} - \mathbb{E}_{P_v}[X]\Big\|_2 \leq \tilde{\epsilon}\Big\},$$

and note that $H(V|\hat{X}, E) \leq \log |S|$. To bound the size of this set $S$, fix some $v_0 \in S$ and consider,

$$S' := \Big\{v \in \{\pm 1\}^d \text{ s.t. } \|v - v_0\|_2 \leq 2\tilde{\epsilon}/(p\tilde{\sigma}_E)\Big\}.$$

Using the fact that $\mathbb{E}_{P_v}[X] = p\tilde{\sigma}_E v$ and the triangle inequality, we have $S \subseteq S'$. To bound the cardinality of $S'$ note that for $v \in S'$, $v$ cannot differ from $v_0$ on more than $N = \lceil \tilde{\epsilon}^2/(p^2\tilde{\sigma}_E^2)\rceil = \lceil d/(16\log d)\rceil$ coordinates. Since $N \leq d/2$,

$$|S'| \leq \sum_{j=1}^{N}\binom{d}{j} \leq N\binom{d}{N} \leq 2^{N\log d + \log N}.$$

Thus for $d \geq 2$,

$$H(V|\hat{X}, E) \leq N \log d + \log(N)$$

$$\leq \left(\frac{d}{16 \log d} + 1\right) \log(d) + \log\left(\frac{d}{16 \log d} + 1\right)$$

$$\leq 3d/4.$$

We conclude that

$$I(V; \hat{X}) \geq d/6.$$

This concludes the proof of Eq. 8.

Proof of Eq. 9: Consider $I(V; X^{(1)}, \ldots, X^{(n)})$. By Lemma 12 and the fact that $X^{(i)}$ and $X^{(j)}$ are identically and independently distributed conditioned on $V$, we have

$$I(V; X^{(1)}, \ldots, X^{(n)}) = \sum_{i=1}^{n} I(V; X^{(i)}) = nI(V; X).$$

Similarly, since the coordinates of $X$ are independent we have

$$I(V; X^{(1)}, \ldots, X^{(n)}) = n \sum_{i=1}^{d} I(V; X_i).$$

Next we give an expression for $I(V; X_i)$ in terms of $p$:

$$I(V; X_i) = H(X_i) - H(X_i|V)$$

$$= 1 - h_2\left(\frac{1}{2} + p\right),$$

where $h_2(t)$ is the binary entropy: $h_2(t) := -t \log(t) - (1-t) \log(1-t)$. We claim

$$h_2\left(\frac{1}{2} + p\right) \geq 1 - 28p^2. \tag{10}$$

Assuming Eq. 10 we have,

$$I(V; X^{(1)}, \ldots, X^{(n)}) = ndI(V; X_i) \leq 28ndp^2.$$

It remains to show that $h_2\left(\frac{1}{2} + p\right) \geq 1 - 28p^2$. Observe the following equality:

$$h_2\left(\frac{1}{2} + p\right) = -\left(\left(\frac{1}{2} + p\right) \log\left(\frac{1}{2} + p\right) + \left(\frac{1}{2} - p\right) \log\left(\frac{1}{2} - p\right)\right)$$

$$= -\left(\left(\frac{1}{2} + p\right) \log\left(\frac{1}{2}(1 + 2p)\right) + \left(\frac{1}{2} - p\right) \log\left(\frac{1}{2}(1 - 2p)\right)\right)$$

$$= -\left(\log\left(\frac{1}{2}\right) + \left(\frac{1}{2} + p\right) \log(1 + 2p) + \left(\frac{1}{2} - p\right) \log(1 - 2p)\right)$$

$$= 1 - \left(\frac{1}{2} \log(1 - 4p^2) + p \log\left(\frac{1 + 2p}{1 - 2p}\right)\right).$$

So it is equivalent to upper bound the following by $28p^2$:

$$\frac{1}{2} \log(1 - 4p^2) + p \log\left(\frac{1 + 2p}{1 - 2p}\right).$$

Since $p \geq 0$, $\frac{1}{2} \log(1 - 4p^2) \leq 0$. Hence, it suffices to show that $\log((1 + 2p)/(1 - 2p)) \leq 28p$. To do this we use the facts that $1/(1 - x) \leq 1 + 2x$ for $x \leq 1/4$, and $\log(1 + x) \leq 2x$ for any $x \geq 0$, and finally $0 \leq p \leq 1/4$:

$$\log\left(\frac{1 + 2p}{1 - 2p}\right) \leq \log((1 + 2p)(1 + 4p))$$

$$\leq \log(1 + 14p)$$

$$\leq 28p.$$

$\square$

## B.2 Proving Theorem 5 and Corollary 2

We establish our lower bound for SCO with sub-exponential noise by establishing a correspondence between Problem 3 and solving Problem 1 given a $\sigma_E$-ESGO. Specifically, for any random variable $X$ in Problem 3 with $\tilde{\epsilon} = 48\epsilon\sqrt{\log d}/R$, we design the following convex function whose optimal point is related to $\mathbb{E}[X]$,

$$\bar{f}(x) := \mathbb{E}[f_X(x)], \tag{11}$$

where

$$f_X(x) := -\frac{1}{3}\langle x, X\rangle + \frac{2L}{3} \cdot \max\left\{0, \|x\| - \frac{R}{2}\right\}. \tag{12}$$

**Lemma 13.** *Denote $w := \mathbb{E}[X]/\|\mathbb{E}[X]\|$. Given that $\mathbb{E}[X] \leq L$, the function $\bar{f}$ defined in (11) has the following properties:*

1. *$\bar{f}$ is convex.*

2. *$\bar{f}$ is minimized at $x^* = \frac{R}{2}w$.*

3. *Every $\epsilon$-optimum $x$ of $\bar{f}$ satisfies*

$$\left(\frac{x}{\|x\|}\right)^\top w \geq 1 - \frac{6\epsilon}{R\mathbb{E}[X]}.$$

4. *For*

$$\hat{g}_X(x) := -\frac{X}{3} + \frac{2L}{3} \cdot \mathbf{1}\left\{\|x\| - \frac{R}{2} > 0\right\} \cdot \frac{x}{\|x\|}, \tag{13}$$

*we have*

$$\mathbb{E}_X[\hat{g}_X(x)] \in \partial\bar{f}(x), \quad \forall x \in \mathbb{R}^d.$$

5. *For any $t > 0$ and unit vector $u \in \mathbb{R}^d$, we have*

$$\mathbb{P}[|\langle u, \hat{g}_X(x) - \nabla f(x)\rangle| \geq t] \leq 2\exp\left(-t\sqrt{d}/\sigma_E\right).$$

*Proof.* Proof of (1): By linearity of expectation,

$$\bar{f}(x) = -\frac{1}{3}\langle x, \mathbb{E}[X]\rangle + \frac{2L}{3} \cdot \max\left\{0, \|x\| - \frac{R}{2}\right\}. \tag{14}$$

This is a convex function of $x$ since it is the sum of a linear function and the max of two convex functions.

Proof of (2): Suppose $\|x\| = 1$. We claim that for any $c > R/2$:

$$\bar{f}(cx) \geq \bar{f}\left(\frac{R}{2}x\right)$$

given that

$$\bar{f}(cx) \geq \bar{f}(Rx/2) \iff -\frac{1}{3}cx^\top\mathbb{E}[X] + \frac{2L}{3}\left(c - \frac{R}{2}\right) \geq -\frac{1}{3} \cdot \frac{R}{2}x^\top\mathbb{E}[X]$$

$$\iff \frac{2L}{3}\left(c - \frac{R}{2}\right) \geq \frac{1}{3}x^\top\mathbb{E}[X]\left(c - \frac{R}{2}\right)$$

$$\iff L \geq \frac{1}{2}x^\top\mathbb{E}[X].$$

Since $\|\mathbb{E}[X]\| \leq L$, the last line holds. Recalling $w = \mathbb{E}[X]/\|\mathbb{E}[X]\|$, notice that if $x$ satisfies $x^\top w \geq 0$, we can decrease the function value by increasing the norm of $x$ up to the value of $R/2$. That is, for any $c < R/2$ we have

$$\bar{f}(cx) > \bar{f}\left(\frac{R}{2}x\right). \tag{15}$$

Therefore, since the optimal $x$ satisfies that $x^\top w \geq 0$, we must have $x^* = (R/2)w$.

Proof of (3): Given any $\epsilon$-optimum $x$, since we must have that $x^\top w \geq 0$, it follows by Eq. 15 that $\frac{R}{2} \cdot \frac{x}{\|x\|}$ is also an $\epsilon$-optimum. Therefore,

$$\bar{f}\left(\frac{R}{2} \cdot \frac{x}{\|x\|}\right) - \bar{f}(x^*) = \frac{1}{3}\left(x^* - \frac{R}{2}\frac{x}{\|x\|}\right)^\top \mathbb{E}[X] \leq \epsilon.$$

Since $x^* = (R/2)w$ we equivalently have

$$\frac{R}{6}\left(w - \frac{x}{\|x\|}\right)^\top \mathbb{E}[X] \leq \epsilon.$$

Using that $\|w\|^2 = 1$ and rearranging terms we get

$$\left(\frac{x}{\|x\|}\right)^\top w \geq 1 - \frac{6\epsilon}{R\|\mathbb{E}[X]\|}.$$

Proof of (4): By Eq. 14 and linearity of expectation we have

$$\mathbb{E}_X[\hat{g}_X(x)] = -\frac{\mathbb{E}[X]}{3} + \frac{2L}{3} \cdot \mathbf{1}\left\{\|x\| - \frac{R}{2} > 0\right\} \cdot \frac{x}{\|x\|} \in \partial\bar{f}(x). \tag{16}$$

Proof of (5): For any $x$, $\hat{g}_X(x)$ takes the form

$$\hat{g}_X(x) = c_1 X + y, \tag{17}$$

where $c_1 = -\frac{1}{3}$ and $y = \frac{2L}{3} \cdot \mathbf{1}\left\{\|x\| - \frac{R}{2} > 0\right\} \cdot \frac{x}{\|x\|}$. Then by Lemma 10 we can conclude that

$$\mathbb{P}[|\langle u, \hat{g}_X(x) - \nabla f(x)\rangle| \geq t] \leq \exp\left(-t\sqrt{d}/\sigma_E\right)$$

for any $t > 0$ and any unit vector $u \in \mathbb{R}^d$. $\qquad\square$

The next lemma establishes a lower bound for Problem 1 with access to actual gradients, or equivalently a $\sigma_E$-ESGO with $\sigma_E = 0$.

**Lemma 14** (Theorem 3 of [24]). *Any algorithm that solves Problem 1 with success probability at least $2/3$ must make at least $\Omega(\min\{RL/\epsilon^2, d\})$ queries.*

**Theorem 5.** *Any algorithm which solves Problem 1 with probability at least $2/3$ using a $\sigma_E$-ESGO makes at least $\tilde{\Omega}(R^2\sigma_E^2/\epsilon^2 + \min\{R^2L^2/\epsilon^2, d\})$ queries.*

*Proof.* Consider the function $\bar{f}(\cdot)$ defined in Eq. 11 with the random variable $X \sim P_v$ defined in Eq. 6, where $v$ is chosen uniformly at random from $\{\pm 1\}^d$. Let $x$ denote the output of the algorithm after making $n$ queries. If $x$ is an $\epsilon$-optimal point, then by Lemma 13,

$$\left(\frac{x}{\|x\|}\right)^\top \left(\frac{\mathbb{E}[X]}{\|\mathbb{E}[X]\|}\right) \geq 1 - \frac{6\epsilon}{R\|\mathbb{E}[X]\|}.$$

Using the fact that, for any unit-norm vectors $u$ and $v$, $\|u - v\|_2 = \sqrt{2(1 - u^\top v)}$, we have

$$\left\|\frac{x}{\|x\|} - \frac{\mathbb{E}[X]}{\|\mathbb{E}[X]\|}\right\|_2 \leq \sqrt{\frac{12\epsilon}{R\|\mathbb{E}[X]\|}}.$$

For $X$ defined in Eq. 6 we know that $\|\mathbb{E}[X]\| = 4\tilde{\epsilon}\sqrt{\log d}$, regardless of the value of $v$. Therefore, defining $\hat{X} := \frac{\|\mathbb{E}[X]\|}{\|x\|} \cdot x$, and set $\epsilon = \frac{\tilde{\epsilon}R}{48\sqrt{\log d}}$, we have

$$\left\|\hat{X} - \mathbb{E}[X]\right\|_2 \le \sqrt{12\epsilon\|\mathbb{E}[X]\|/R} \le \tilde{\epsilon}. \tag{18}$$

Next, we observe that we can simulate the responses of the subgradient oracle defined in Eq. 13 using only $n$ i.i.d. samples of $X \sim P_v$, which is a $\sigma_E$-ESGO by Lemma 13. Therefore, any algorithm which can output an $\epsilon$-optimal point $x$ with probability at least $2/3$ can be used to construct a mean-estimation algorithm which outputs an estimate $\hat{X}$ that satisfies $\|\hat{X} - \mathbb{E}[X]\| \le \tilde{\epsilon}$ with probability at least $2/3$. Therefore by the hardness of mean estimation with isotropic noise, established in Lemma 11, we must have that

$$n \ge \Omega\left(\frac{\sigma_E^2}{\tilde{\epsilon}^2 \log d}\right) = \tilde{\Omega}(\sigma_E^2 R^2/\epsilon^2).$$

Combined with Lemma 14, we obtain the desired result. $\qquad\square$

Finally, we obtain our lower bound for Problem 1 given a $(\sigma_I, \delta)$-ISGO as a simple corollary.

**Corollary 2.** *Any algorithm which solves Problem 1 with probability at least $2/3$ using a $(\sigma_I, \delta)$-ISGO makes at least $\tilde{\Omega}(R^2\sigma_I^2/(\epsilon^2\log^2(1/\delta)) + \min\{R^2L^2/\epsilon^2, d\})$ queries.*

*Proof.* By Lemma 8, any $\sigma_E$-ESGO is a $(\sigma_E\log(2/\delta), \delta)$-ISGO. The result then follows by setting $\sigma_E \leftarrow \sigma_I/\log(2/\delta)$ and applying Theorem 5. $\qquad\square$

## C   Quantum isotropifier and an improved bound for quantum SCO

### C.1   Qubit notation and conventions.

We use the notation $|\cdot\rangle$ to denote input or output registers composed of qubits that can exist in *superpositions*. Specifically, given $m$ points $x_1, \ldots, x_m \in \mathbb{R}^d$ and a coefficient vector $\mathbf{c} \in \mathbb{C}^m$ such that $\sum_{i\in[m]} |c_i|^2 = 1$, the quantum register could be in the state $|\psi\rangle = \sum_{i\in[m]} c_i |x_i\rangle$, which represents a superposition over all $m$ points simultaneously. Upon measuring this state, the outcome will be $x_i$ with probability $|c_i|^2$. Moreover, to characterize a classical probability distribution $p$ over $\mathbb{R}^d$ in a quantum framework, we can prepare the quantum state $\int_{x\in\mathbb{R}^d} \sqrt{p(x)\mathrm{d}x} |x\rangle$, which we denote as the state over $\mathbb{R}^d$ with wave function $\sqrt{p(x)}$. Measuring this state will yield outcomes according to the probability density function $p$. When applicable, we use $|\mathrm{garbage}(\cdot)\rangle$ to denote possible garbage states.[10]

Throughout this paper, we assume that any quantum oracle $\mathcal{O}$ is a unitary operation, and we can also access its inverse $\mathcal{O}^{-1}$ satisfying $\mathcal{O}^{-1}\mathcal{O} = \mathcal{O}\mathcal{O}^{-1} = I$. This is a standard assumption in prior works on quantum algorithms, see e.g. [22, 53].

### C.2   Bounded random variables

In this subsection, we introduce our quantum multivariate mean estimation algorithm for bounded random variables whose error is small in any direction with high probability. This algorithm is a variant of [22, Algorithm 2]. We begin by presenting some useful algorithmic components.

**Lemma 15** (Directional mean oracle, Proposition 3.2 of [22])**.** *Suppose we have access to the quantum sampling oracle $\mathcal{O}_X$ of a bounded random variable $X$ satisfying $\|X\| \le 1$. Then for any $\nu > 0$, there exists two procedures,* QDirectionalMean1$(X, m, \alpha, \nu)$ *and* QDirectionalMean2$(X, m, \alpha, \nu)$,

---

[10]The garbage state is the quantum counterpart of classical garbage information generated when preparing a classical random sample or a classical stochastic gradient, which, in general, cannot be erased or uncomputed. In this work, we consider a general model without any assumptions about the garbage state. See e.g., [27, 53] for a similar discussion on the standard use of garbage quantum states.

Throughout this paper, whenever we query a quantum oracle that contains a garbage state, we do not assume we know its identity. Nevertheless, our algorithm requires that the garbage state be maintained coherently as part of the system to perform the inverse operation.

*that respectively uses $\tilde{O}(m\sqrt{\mathbb{E}|X|}\log^2(1/\nu))$ and $\tilde{O}(m\log^2(1/\nu))$ queries, and output quantum states*

$$|\psi_{\text{out},1}\rangle = |\psi_{\text{prod}}\rangle + |\perp_1\rangle, \qquad |\psi_{\text{out},2}\rangle = |\psi_{\text{prod}}\rangle + |\perp_2\rangle,$$

*where*

$$|\psi_{\text{prod}}\rangle = \bigotimes_{j=1}^{d} \Big( \frac{1}{\sqrt{m}} \sum_{g_j \in G_m} e^{im\alpha g_j \mathbb{E}[X_j]} |g_j\rangle \Big),$$

*and*

$$\begin{cases} \||\perp_1\rangle\| \le \nu + \sqrt{\underset{g\sim G_m^d}{\mathbb{P}}[\alpha\mathbb{E}|\langle g,X\rangle| > \mathbb{E}\|X\|]} + 2\sqrt{m\alpha}d^{1/4}e^{-1/\alpha^2}, \\ \||\perp_2\rangle\| \le \nu + \sqrt{\underset{g\sim G_m^d}{\mathbb{P}}[\alpha\mathbb{E}|\langle g,X\rangle| > 1]} + 2\sqrt{m\alpha}d^{1/4}e^{-1/\alpha^2}. \end{cases}$$

Using $\texttt{QDirectionalMean}\beta(\cdot)$ as a subroutine, [22] develops a quantum multivariate mean estimation algorithms for bounded random variables. In this work, we provide an improved error analysis of this algorithm, based on which we apply the boosted unbiased phase estimation technique introduced in [57] to suppress the bias in multivariate mean estimation.

**Lemma 16** (Boosted Unbiased Phase Estimation, Theorem 28 of [57])**.** *Given $k$ copies of a quantum state $|\psi\rangle = \frac{1}{\sqrt{m}}\sum_{g\in G_m} e^{im\phi g}|g\rangle$ for some unknown phase $\phi$ satisfying $-\frac{2\pi}{3} \le \phi \le \frac{2\pi}{3}$, there is a procedure $\texttt{BoostedUnbiasedPhaseEstimation}(|\psi\rangle^{\otimes k})$ that returns an unbiased estimate $\hat{\phi}$ satisfying*

$$\mathbb{P}[|\hat{\phi} - \phi| \le 6/m] \ge 1 - e^{-k}.$$

**Proposition 2.** *For any bounded random variable $X$ satisfying $\|X\| \le 1$ and $\min_{X\neq 0}\|X\| \ge \epsilon$, Algorithm 2 and its output $\hat{\mu}$ satisfy the following:*

1. *$\|\hat{\mu}\| \le \sqrt{d}$.*

2. *$\hat{\mu}$ is almost unbiased, i.e., $\|\mathbb{E}[\hat{\mu}] - \mathbb{E}[X]\| \le \tilde{O}(\delta\sqrt{d})$.*

3. *For any coordinate $j \in [d]$, we have*

$$\mathbb{P}\big[|\hat{\mu}_j - \mathbb{E}[X_j]| \ge \epsilon\big] \le \tilde{O}(\delta).$$

4. *For any unit vector $u \in \mathbb{R}^d$, we have*

$$\mathbb{P}\big[|\langle u, \hat{\mu} - \mathbb{E}[X]\rangle| \ge \epsilon\log(1/\delta)\big] \le \tilde{O}(\delta d).$$

5. *Algorithm 2 uses $\tilde{O}\big(\sqrt{\mathbb{E}\|X\|}\log^3(1/\delta)/\epsilon\big)$ queries to $\mathcal{O}_X$ when $\beta = 1$ and $\tilde{O}\big(\log^3(1/\delta)/\epsilon\big)$ queries when $\beta = 2$.*

*Proof.* By Lemma 15, the quantum state $|\psi_{\text{out}}\rangle$ in Line 4 is defined on $G_m^d$. Hence, we have $\|\hat{\mu}\|_\infty \le 1$ and $\|\hat{\mu}\| \le \sqrt{d}$. Moreover, it satisfies

$$|\psi_{\text{out}}\rangle = |\psi_{\text{prod}}\rangle + |\perp_\beta\rangle, \tag{19}$$

where

$$|\psi_{\text{prod}}\rangle = \bigotimes_{j=1}^{d} \Big( \frac{1}{\sqrt{m}} \sum_{g_j \in G_m} e^{im\alpha g_j \mathbb{E}[X_j]} |g_j\rangle \Big)$$

We first analyze the algorithm with $|\psi_{\text{out}}\rangle$ replaced by $|\psi_{\text{prod}}\rangle$ in Line 4. Note that $|\psi_{\text{prod}}\rangle$ is a product state, so the outcomes $\hat{\phi}_1, \ldots, \hat{\phi}_d$ of $\texttt{BoostedUnbiasedPhaseEstimation}$ follows a

**Algorithm 2:** Bounded quantum mean estimation with boosted unbiased phase estimation (`Debiased-QBounded`)

---

**Input:** Random variable $X$, target accuracy $0 < \epsilon \leq 1$, failure probability $\delta \leq \frac{1}{2}$, choice of subroutines $\beta = 1$ or $2$

**Output:** An estimate $\hat{\mu}$ of $\mathbb{E}[X]$

1 Set $\epsilon' \leftarrow \epsilon/8, n \leftarrow \sqrt{\mathbb{E}\|X\|} \log(d/\delta)/\epsilon'$

2 Set $\alpha \leftarrow \frac{1}{\sqrt{\log(1000\pi n\sqrt{d})}}, m = 2^{\left\lceil \log\left(\frac{12\pi}{\alpha\epsilon'}\right)\right\rceil}$, and $\nu \leftarrow \frac{\delta}{9}$

3 **for** $k = 1, \ldots, \lceil 18 \log(1/\delta)\rceil$ **do**

4 $\quad |\psi_{\text{out}}^{(k)}\rangle \leftarrow \text{QDirectionalMean}\beta(X, m, \alpha, \nu)$

5 Run `BoostedUnbiasedPhaseEstimation` on all the $d$ coordinates independently and denote $\hat{\phi}_1, \ldots, \hat{\phi}_d$ to be the outcomes

6 **return** $\hat{\mu} \leftarrow \frac{2\pi}{\alpha}(\hat{\phi}_1, \ldots, \hat{\phi}_d)^\top$

---

product distribution. Hence, in this ideal case $\hat{\mu}$ also follows a product distribution, which we denote as

$$\hat{p}_1 \otimes \hat{p}_2 \otimes \cdots \otimes \hat{p}_d.$$

Given that the phase in $|\psi_{\text{prod}}\rangle$ satisfies $|\alpha\mathbb{E}[X]|_\infty \leq 2\pi/3$, by Lemma 16, we have

$$\mathbb{E}_{\hat{\mu}\sim\hat{p}_1\otimes\cdots\otimes\hat{p}_d}[\hat{\mu}] - \mathbb{E}[X] = 0,$$

and

$$\mathbb{P}\left[\left|\hat{\phi}_j - \frac{\alpha}{2\pi}\mathbb{E}[X_j]\right| \geq \frac{6}{m}\right] \leq \delta, \quad \mathbb{P}_{\hat{\mu}_j\sim\hat{p}_j}\left[|\hat{\mu}_j - \mathbb{E}[X_j]| \geq \epsilon\right] \leq \delta,$$

Thus, for any unit vector $u \in \mathbb{R}^d$, the random variable

$$\langle u, \hat{\mu} - \mathbb{E}[X]\rangle, \quad \hat{\mu} \sim \hat{p}_1 \otimes \cdots \otimes \hat{p}_d$$

satisfies

$$\mathbb{P}_{\hat{\mu}\sim\hat{p}_1\otimes\cdots\otimes\hat{p}_d}\left[|\langle u, \hat{\mu} - \mathbb{E}[X]\rangle| \geq \epsilon\log(1/\delta)\right] \leq O(\delta d)$$

by Lemma 22 where we set $k = d$.

Next, we discuss the error caused by the difference between $|\psi_{\text{prod}}\rangle$ and the actual state $|\psi_{\text{out}}\rangle$, i.e., the $|\perp_\beta\rangle$ term in Eq. 19. By Lemma 15, we have

$$\||\perp_1\rangle\| \leq \nu + \sqrt{\mathbb{P}_{g\sim G_m^d}[\alpha\mathbb{E}|\langle g, X\rangle| > \mathbb{E}\|X\|]} + 2\sqrt{m}\alpha d^{1/4}e^{-1/\alpha^2}$$

$$\leq \frac{\delta}{3} + \sqrt{\mathbb{P}_{g\sim G_m^d}[\alpha\mathbb{E}|\langle g, X\rangle| > \mathbb{E}\|X\|]}$$

and

$$\||\perp_2\rangle\| \leq \nu + \sqrt{\mathbb{P}_{g\sim G_m^d}[\alpha\mathbb{E}|\langle g, X\rangle| > 1]} + 2\sqrt{m}\alpha d^{1/4}e^{-1/\alpha^2}$$

$$\leq \frac{\delta}{3} + \sqrt{\mathbb{P}_{g\sim G_m^d}[\alpha\mathbb{E}|\langle g, X\rangle| > 1]}$$

given the choice of parameters of $\alpha, m, \gamma$ in Algorithm 2. By Corollary 9, we have

$$\mathbb{P}_{g\sim G_m^d}[\alpha\mathbb{E}|\langle g, X\rangle| > \mathbb{E}\|X\|] \leq 16\alpha\sqrt{d}e^{-1/(32\alpha^2)}\log\left(\alpha\sqrt{d}/\epsilon\right) \leq \frac{\delta^2}{36},$$

and

$$\mathbb{P}_{g\sim G_m^d}\big[\alpha\mathbb{E}|\langle g, X\rangle| > 1\big] \le 4\alpha\sqrt{d}e^{-1/(2\alpha^2)} \le \frac{\delta^2}{36}.$$

which leads to $\|\,|\bot\rangle_1\,\|, \|\,|\bot\rangle_2\,\| \le \delta/2$. Hence, the actual probability distribution $\hat{p}$ of $\hat{\mu}$ satisfies

$$\big\|\hat{p} - \hat{p}_1\otimes\hat{p}_2\otimes\cdots\otimes\hat{p}_d\big\|_1 \le 2\lceil 18\log(1/\delta)\rceil\|\,|\psi_{\text{out}}\rangle - |\psi_{\text{prod}}\rangle\,\| = 72\delta\log(1/\delta).$$

Then, we can derive that

$$\Big\|\mathbb{E}_{\hat{\mu}\sim\hat{p}}[\hat{\mu}] - \mathbb{E}[X]\Big\| \le \tilde{O}\big(\delta\sqrt{d}\big),$$

given that $\|\hat{\mu}\| \le \sqrt{d}$ and $\|X\| \le 1$. Moreover, we have

$$\mathbb{P}_{\hat{\mu}\sim\hat{p}}\big[|\hat{\mu}_j - \mathbb{E}[X_j]| \ge \epsilon\big] \le \delta + 72\delta\log(1/\delta) = \tilde{O}(\delta),$$

and

$$\mathbb{P}_{\hat{\mu}\sim\hat{p}}\big[|\langle u, \hat{\mu} - \mathbb{E}[X]\rangle| \ge \epsilon\big] \le \tilde{O}(\delta d) + 72\delta\log(1/\delta) = \tilde{O}(\delta d).$$

By Lemma 15, when $\beta = 1$ the number of queries to $\mathcal{O}_X$ is

$$\lceil 18\log(1/\delta)\rceil \cdot \tilde{O}\big(m\sqrt{\mathbb{E}\|X\|}\log^2(1/\nu)\big) = \tilde{O}\big(\sqrt{\mathbb{E}\|X\|}\log^3(1/\delta)/\epsilon\big).$$

For $\beta = 2$, the number of queries is

$$\lceil 18\log(1/\delta)\rceil \cdot \tilde{O}\big(m\log^2(1/\nu)\big) = \tilde{O}\big(\log^3(1/\delta)/\epsilon\big). \tag{20}$$

$\square$

### C.3 Unbounded random variables with bounded expectation

In this subsection, we introduce our quantum multivariate mean estimation algorithm for unbounded random variable with bounded expectation, a variant of [22, Algorithm 2], obtained by applying Algorithm 2 to a series of truncated bounded random variables.

---

**Algorithm 3:** Unbounded quantum mean estimation (`QUnbounded`)

---

**Input:** Random variable $X$, target accuracy $0 < \epsilon \le 1$, failure probability $\delta$
**Output:** An estimate $\hat{\mu}$ of $\mathbb{E}[X]$ with $\ell_\infty$ error at most $\epsilon$

1 Set $\sigma' \leftarrow \sigma/\log(\sigma/\epsilon)$, $\epsilon' \leftarrow \epsilon/(\sigma'\log(1/\delta))$, $K \leftarrow \lceil 2\log\big(2\sqrt{2}/\epsilon'\big)\rceil$
2 Take $\lceil 64\log^2(1/\epsilon)\log(d/\delta)\rceil$ classical random samples $X_1,\ldots,X_{\lceil 64\log^2(1/\epsilon)\log(d/\delta)\rceil}$, use $\eta$ to
   denote their coordinate median
3 Define a new random variable $Y \leftarrow X - \eta$
4 $a_{-1} \leftarrow 0$
5 **for** $k = 0,\ldots,K$ **do**
6    $a_k \leftarrow 2^k\sigma'$
7    Define the bounded random variable $Y_k := \frac{Y}{a_k}\cdot\mathbb{I}\{a_{k-1} \le \|Y\| < a_k\}$
8    **if** $k = 0$ **then** $\hat{\mu}'_k \leftarrow$ `Debiased-QBounded2`$(Y_k, 2^{-k-1}\epsilon'/K, \delta/(Kd))$
9    **else** $\hat{\mu}'_k \leftarrow$ `Debiased-QBounded1`$(Y_k, 2^{-k-1}\epsilon'/K, \delta/(Kd))$
10    **if** $\|\hat{\mu}'_k\| \le 2^{-2k+2}$ **then** $\hat{\mu}_k \leftarrow \hat{\mu}'_k$
11    **else** $\hat{\mu}_k \leftarrow 0$
12 **return** $\hat{\mu} \leftarrow \eta + \sum_{k=0}^K a_k\hat{\mu}_k$

---

**Lemma 17** (Theorem 2 of [44]). *For any $n$ independent samples $X_1,\ldots,X_n$ of a random variable $X \in \mathbb{R}^d$, any $\delta > 0$, and any $n \ge \lceil 32\log(d/\delta)\rceil$, their coordinate-wise median $\eta$ satisfies*

$$\mathbb{P}\left[\|\eta - \mathbb{E}[X]\| \le \sigma\sqrt{\frac{32\log(d/\delta)}{n}}\right] \ge 1 - \delta. \tag{21}$$

**Proposition 3.** *For any $\epsilon, \delta > 0$ and any random variable $X \in \mathbb{R}^d$ with variance $\mathrm{Var}[X] \leq \sigma^2$, Algorithm 3 outputs an estimate $\hat{\mu}$ satisfying $\mathbb{E}[\hat{\mu}] - \mathbb{E}[X] \leq \sigma/\log(1/\epsilon)$ and*

$$\mathbb{P}\big[\big|\langle u, \hat{\mu} - \mathbb{E}[X]\rangle\big| \geq \epsilon\big] \leq \tilde{O}(\delta) \tag{22}$$

*for any unit vector $u \in \mathbb{R}^d$. Moreover, Algorithm 3 makes $\tilde{O}\big(\sigma \log^5(1/\delta)/\epsilon\big)$ queries to $\mathcal{O}_X$.*

*Proof.* Denote $\hat{\mu}_Y = \sum_{k=0}^{K} a_k \hat{\mu}_k$. We first consider the case that $\|X - \eta\| \leq \sigma'$, which happens with probability at least $1 - \delta$ by Lemma 17. Under this condition, we have

$$\mathbb{P}\left[\big|\langle u, \hat{\mu}_k - \mathbb{E}[Y_k]\rangle\big| > \frac{2^{-k-1}\epsilon}{K} \,\Big|\, \|X - \eta\| \leq \sigma'\right] \leq \tilde{O}(\delta/K), \qquad \forall k = 0, \ldots, K, \tag{23}$$

by Proposition 2. Additionally, we have

$$\mathbb{E}[\|Y\|^2] \leq \|\eta - \mathbb{E}[X]\|^2 + \mathbb{E}[\|X - \mathbb{E}[X]\|^2] \leq \sigma^2 + \mathrm{Var}[X] = 2\sigma^2,$$

and

$$\mathbb{P}\{\|Y\| \geq a_K\} \leq \frac{\mathbb{E}[\|Y\|^2]}{a_K^2} \leq 2^{-2K+1}.$$

Therefore,

$$\mathbb{E}\|Y_k\| \leq \mathbb{P}[a_{k-1} \leq \|Y\| \leq a_k] \leq \mathbb{P}[\|Y\| \geq a_{k-1}] \leq \frac{1}{2^{2k+1}}, \qquad \forall k = 0, \ldots, K.$$

From this, we deduce that if

$$|\langle u, \hat{\mu}_k - \mathbb{E}[Y_k]\rangle| > \frac{2^{-k-1}\epsilon}{K} \tag{24}$$

holds for all unit vectors $u \in \mathbb{R}^d$, we have $|\hat{\mu}_k| \leq 2^{-2k+2}$ and $\hat{\mu}_k' = \hat{\mu}_k$. Consequently,

$$\mathbb{P}\left[\Big|\Big\langle u, \hat{\mu}_Y - \sum_{k=0}^{K} a_k \mathbb{E}[Y_k]\Big\rangle\Big| \geq \frac{\epsilon}{2} \,\Big|\, \|X - \eta\| \leq \sigma'\right] \leq \tilde{O}(\delta)$$

by union bound. Furthermore, we have

$$\left\|\mathbb{E}[Y] - \sum_{k=0}^{K} a_k \mathbb{E}[Y_k]\right\| = \big\|\mathbb{E}[Y \cdot \mathbb{I}\{\|Y\| \geq a_K\}]\big\|$$

$$\leq \mathbb{E}[\|Y\| \cdot \mathbb{I}\{\|Y\| \geq a_K\}]$$

$$\leq \sqrt{\mathbb{E}[\|Y\|]^2 \cdot \mathbb{P}\{\|Y\| \geq a_K\}} \leq \frac{\epsilon}{2}$$

which leads to

$$\mathbb{P}\big[\big|\langle u, \hat{\mu}_Y - \mathbb{E}[Y]\rangle\big| \geq \epsilon \,\big|\, \|X - \eta\| \leq \sigma'\big] \leq \tilde{O}(\delta).$$

Note that $\|\eta\| \leq \|\mathbb{E}[X]\| + \sigma \leq L + \sigma$ in this case, we have $\hat{\mu} - \mathbb{E}[X] = \hat{\mu}_Y - \mathbb{E}[Y]$. Thus,

$$\mathbb{P}\big[\big|\langle u, \hat{\mu} - \mathbb{E}[X]\rangle\big| \geq \epsilon \,\big|\, \|X - \eta\| \leq \sigma'\big] \leq \tilde{O}(\delta).$$

Counting in the error probability when $\|\eta - \mathbb{E}[X]\| \geq \sigma'$, we have

$$\mathbb{P}\big[\big|\langle u, \hat{\mu} - \mathbb{E}[X]\rangle\big| \geq \epsilon\big] \leq \tilde{O}(\delta) + \delta = \tilde{O}(\delta). \tag{25}$$

As for the bias of $\hat{\mu}$, we have

$$\big\|\mathbb{E}[\hat{\mu}] - \mathbb{E}[X]\big\| \leq \sum_{k=0}^{K} a_k \big\|\mathbb{E}[\hat{\mu}_k]\big\| + \big\|\mathbb{E}[\eta] - \mathbb{E}[X]\big\| = \sum_{k=0}^{K} a_k \big\|\mathbb{E}[\hat{\mu}_k]\big\| \leq \frac{\sigma}{\log(\sigma/\epsilon)}$$

Next, we discuss the query complexity of Algorithm 3. The number of queries in the iteration $k = 0$ is

$$\tilde{O}\Big(\frac{K \log^3(5K/\delta)}{\epsilon'}\Big) = \tilde{O}\Big(\frac{\sigma \log^4(1/\delta)}{\epsilon}\Big),$$

and the number of queries in the $k$-th iteration for $k > 0$ is

$$\tilde{O}\Big(\frac{K\sqrt{\mathbb{E}\|Y_k\|}\log^3(1/\delta)}{2^{-k-1}\epsilon'}\Big) = \tilde{O}\Big(\frac{\sigma\log^4(1/\delta)}{\epsilon}\Big).$$

Combining with the number of classical samples to obtain $\mu$, we can conclude that the total number of queries equals

$$\tilde{O}\Big(\frac{\sigma\log^4(1/\delta)}{\epsilon}\Big) + K \cdot \tilde{O}\Big(\frac{\sigma\log^4(1/\delta)}{\epsilon}\Big) = \tilde{O}\Big(\frac{\sigma\log^5(1/\delta)}{\epsilon}\Big).$$

$\square$

## C.4  Removing the bias

In this subsection, we combine Algorithm 3 with the multi-level Monte Carlo (MLMC) technique to obtain an unbiased estimate of an unbounded random variable whose error is small in any direction with high probability.

---

**Algorithm 4:** Quantum Isotropifier

---

**Input:** Random variable $X$, target accuracy $\epsilon$, failure probability $\delta$
**Output:** An unbiased estimate $\hat{\mu}$ of $\mathbb{E}[X]$

1 Define $\beta_j := 2^{-j}j^2, \forall j \in \mathbb{N}$
2 Set $\hat{\mu}^{(0)} \leftarrow \texttt{QUnbounded}(X, \epsilon/6, \delta)$
3 Randomly sample $j \sim \text{Geom}\left(\frac{1}{2}\right) \in \mathbb{N}$
4 $\hat{\mu}^{(j)} \leftarrow \texttt{QUnbounded}(X, \beta_j\epsilon/6, \delta)$
5 $\hat{\mu}^{(j-1)} \leftarrow \texttt{QUnbounded}(X, \beta_{j-1}\epsilon/6, \delta)$
6 $\hat{\mu} \leftarrow \hat{\mu}^{(0)} + 2^j(\hat{\mu}^{(j)} - \hat{\mu}^{(j-1)})$
7 **return** $\hat{\mu}$

---

**Theorem 8.** *For any $\epsilon, \delta > 0$ and any random variable $X \in \mathbb{R}^d$ with variance $\text{Var}[X] \leq \sigma^2$, the output $\hat{\mu}$ of Algorithm 4 satisfies $\mathbb{E}[\hat{\mu}] - \mathbb{E}[X] = 0$ and*

$$\mathbb{P}\big[|\langle u, \hat{\mu} - \mathbb{E}[X]\rangle| \geq \epsilon\log^2(8/\delta)\big] \leq \tilde{O}(\delta). \tag{26}$$

*for any unit vector $u \in \mathbb{R}^d$. Moreover, Algorithm 4 makes $\tilde{O}\big(\sigma\log^5(1/\delta)/\epsilon\big)$ queries to $\mathcal{O}_X$ in expectation.*

*Proof.* The structure of our proof is similar to the proof of Theorem 4 of [53]. Note that the output $\hat{\mu}$ of Algorithm 4 can be written as

$$\hat{\mu} = \hat{\mu}^{(0)} + 2^J(\hat{\mu}^{(J)} - \hat{\mu}^{(J-1)}), \qquad J \sim \text{Geom}\Big(\frac{1}{2}\Big) \in \mathbb{N}. \tag{27}$$

Thus,

$$\mathbb{E}[\hat{\mu}] = \mathbb{E}[\hat{\mu}^{(0)}] + \sum_{j=1}^{\infty}\mathbb{P}\{J = j\}2^j(\mathbb{E}[\hat{\mu}^{(j)}] - \mathbb{E}[\hat{\mu}^{(j)}]) = \mathbb{E}[\hat{\mu}_\infty] = \mathbb{E}[X].$$

For each $j \in \mathbb{N}$ and $\hat{\mu}^{(j)}$, we denote $\xi^{(j)} = \hat{\mu}^{(j)} - \mathbb{E}[X]$. Then,

$$\mathbb{P}\big[|\langle u, \hat{\mu} - \mathbb{E}[X]\rangle| \geq \epsilon\log^2(8/\delta)\big]$$
$$= \mathbb{P}\big[|\langle u, \xi^{(0)} + 2^J(\xi^{(J)} - \xi^{(J-1)})\rangle| \geq \epsilon\log^2(8/\delta)\big]$$
$$\geq \mathbb{P}\big[|\langle u, \xi^{(0)}\rangle| \geq \epsilon\log^2(8/\delta)/6\big] + \mathbb{P}\big[|\langle u, \xi^{(J)}\rangle| \geq 2^{-J}\epsilon\log^2(8/\delta)/6\big]$$
$$+ \mathbb{P}\big[|\langle u, \xi^{(J-1)}\rangle| \geq 2^{-J}\epsilon\log^2(8/\delta)/6\big],$$

where by Proposition 3 we have

$$\mathbb{P}\big[|\langle u, \xi^{(0)}\rangle| \geq \epsilon\log^2(8/\delta)/6\big] \leq \mathbb{P}\big[|\langle u, \xi^{(0)}\rangle| \geq \epsilon/6\big] \leq \tilde{O}(\delta),$$

and

$$\mathbb{P}\big[|\langle u, \xi^{(J)}\rangle| \geq 2^{-J}\epsilon \log^2(8/\delta)/6\big] \leq \mathbb{P}\big[|\langle u, \xi^{(J)}\rangle| \geq \beta_J \epsilon/6\big] + \mathbb{P}\big[2^J \beta_J \geq \log^2(8/\delta)\big] = \tilde{O}(\delta).$$

Similarly, we have

$$\mathbb{P}\big[|\langle u, \xi^{(J-1)}\rangle| \geq 2^{-J}\epsilon \log^2(8/\delta)/6\big] \leq \mathbb{P}\big[|\langle u, \xi^{(J)}\rangle| \geq \beta_{J-1}\epsilon/6\big] + \mathbb{P}\big[2^J \beta_{J-1} \geq \log^2(8/\delta)\big] = \tilde{O}(\delta),$$

which gives $\mathbb{P}\big[|\langle u, \hat{\mu} - \mathbb{E}[X]\rangle| \geq \epsilon \log^2(8/\delta)\big] \leq \tilde{O}(\delta)$. Moreover, the number of queries of Algorithm 4 equals

$$\tilde{O}\Big(\frac{\sigma \log^5(1/\delta)}{\epsilon}\Big) \cdot \Big(1 + \sum_{j=1}^{\infty} \mathbb{P}\{J = j\}(\beta_j^{-1} + \beta_{j-1}^{-1})\Big) = \tilde{O}\Big(\frac{\sigma \log^5(1/\delta)}{\epsilon}\Big) \sum_{j=1}^{\infty} \frac{1}{j^2}$$

$$= \tilde{O}\Big(\frac{\sigma \log^5(1/\delta)}{\epsilon}\Big)$$

by Proposition 3. $\qquad\square$

### C.5 An improved bound for quantum SCO

In this subsection, we apply Algorithm 4 to obtain an ISGO using queries to a QVSGO, and then solve SCO using the stochastic cutting plane method developed in Section A.

**Theorem 6.** *For any differentiable $f \colon \mathbb{R}^d \to \mathbb{R}$, a $(\sigma_I, \delta)$-ISGO of $f$ can be implemented using $\tilde{O}(\sigma_V \sqrt{d} \log^7(1/\delta)/\sigma_I)$ queries to a $\sigma_V$-QVSGO.*

*Proof.* For any $x \in \mathbb{R}^d$, it suffices to apply `QuantumIsotropifier` (Algorithm 4) to the QVSGO at $x$. In particular, by Theorem 8, there exists some $\hat{\delta} = \tilde{O}(\delta)$ such that the output of `QuantumIsotropifier`$(\sigma_I d^{-1/2}/\log^2(8/\hat{\delta}), \hat{\delta})$ is an $(\sigma_I, \delta)$-ISGO, and the number of queries equals $\tilde{O}(\sigma_V \sqrt{d} \log^7(1/\hat{\delta})/\sigma_I) = \tilde{O}(\sigma_V \sqrt{d} \log^7(1/\delta)/\sigma_I)$ in expectation. $\qquad\square$

**Theorem 7.** *With success probability at least $2/3$, Problem 2 can be solved using $\tilde{O}(dR\sigma_V/\epsilon)$ queries to a $\sigma_V$-QVSGO.*

*Proof.* The proof is established by combining Theorem 1 and Theorem 6, where we set $\sigma_I = \epsilon\sqrt{d}$. $\qquad\square$

## D  Sub-exponential distributions and additional discussion of SGO oracles

In this section, we review sub-exponential distributions and also further discuss the relationships between the various SGOs we define.

**Review of sub-exponential distributions.**  As there are several equivalent ways to define a sub-exponential random variable [61, Proposition 2.7.1], we will use the following "tail-inequality" version which suits our purposes:

**Definition 11.** *A random variable $X \in \mathbb{R}$ is $\sigma$-sub-exponential if*

$$\mathbb{P}[|X - \mathbb{E}X| \geq t] \leq 2\exp(-t/\sigma) \ \text{ for all } t \geq 0.$$

Analogously to the definition of a sub-Gaussian random vector (see, e.g., Definition 3.4.1 in [61] or Definition 2 in [35]), we say a random vector $X \in \mathbb{R}^d$ is sub-exponential if all of the one-dimensional marginals are sub-exponential random variables:

**Definition 12.** *A random vector $X \in \mathbb{R}^d$ is $\sigma$-sub-exponential if for any unit vector $u \in \mathbb{R}^d$, we have that $\langle X, u \rangle$ is $\sigma$-sub-exponential, namely:*

$$\mathbb{P}[|\langle X - \mathbb{E}X, u \rangle| \geq t] \leq 2\exp(-t/\sigma) \ \text{ for all } t \geq 0.$$

It is well known that sub-exponential distributions generalize sub-Gaussian distributions and therefore bounded random variables in particular. Finally, we prove a short lemma which we will reference below:

**Lemma 18.** *If $X \in \mathbb{R}^d$ is $\sigma$-sub-exponential, then $\mathbb{E}\|X - \mathbb{E}X\|^2 \leq Cd\sigma^2$ for some absolute constant $C$.*

*Proof.* Letting $e_i$ denote the $i$-th standard basis vector, we have

$$\mathbb{E}\|X - \mathbb{E}X\|^2 = \mathbb{E}\sum_{i \in [d]} \langle X - \mathbb{E}X, e_i \rangle^2 = \sum_{i \in [d]} \mathbb{E}\langle X - \mathbb{E}X, e_i \rangle^2 \leq Cd\sigma^2,$$

where the last equality follows because if a random variable $Z \in \mathbb{R}$ is $\sigma$-sub-exponential, then $\mathbb{E}(Z - \mathbb{E}Z)^2 \leq C\sigma^2$ [61, Proposition 2.7.1]. $\quad\square$

**Additional discussion of SGO oracles.** Note that a $\sigma_E$-ESGO as defined in Definition 5 is $\sigma_E/\sqrt{d}$-sub-exponential per Definition 5 as opposed to $\sigma_E$-sub-exponential. We perform this scaling so that, per Lemma 18, a $\sigma_E$-ESGO is also a $C\sigma_E$-VSGO for some absolute constant $C$. In other words, this scaling makes it so that Definition 5 is truly a restriction of Definition 3, and thus the rates of Corollaries 3 and 4 are comparable.

# E   Technical lemmas

In this section, we collect some miscellaneous technical lemmas. The following lemma from [22] shows that for any fixed vector $x \in \mathbb{R}^d$, most of the vectors $g \sim G_m^d$ have a relatively small inner product with $x$.

**Lemma 19** (Lemma 3.1 of [22]). *Let $\alpha > 0$. For any vector $x \in \mathbb{R}^d$ we have*

$$\mathbb{P}_{g \sim G_m^d}[\alpha|\langle g, x \rangle| \geq \|x\|] \leq 2e^{-2/\alpha^2}, \quad \forall x \in \mathbb{R}^d.$$

We extend this result and show that this exponentially small probability bound still holds when $x$ is a random variable instead of a fixed vector.

**Lemma 20.** *Let $\alpha > 0$. Consider a $d$-dimensional random variable $Y \in \mathbb{R}^d$ that satisfies*

$$\mathbb{P}_Y[\alpha|\langle Y, x \rangle| \geq \|x\|] \leq p(\alpha), \quad \forall x \in \mathbb{R}^d, \tag{28}$$

*for some $p(\cdot)$ that is a function of $\alpha$, then for any random variable $X \in \mathbb{R}^d$, we have*

$$\mathbb{P}_Y[\alpha\mathbb{E}_X[|\langle Y, X \rangle|] \geq 2\max\|X\|] \leq \alpha p(\alpha)\max\|Y\| \tag{29}$$

*and*

$$\mathbb{P}_Y[\alpha\mathbb{E}_X|\langle Y, X \rangle| > \mathbb{E}\|X\|] \leq 8\alpha p(8\alpha)\max\|Y\|\log\left(\frac{\alpha\max\|Y\|\sqrt{\mathbb{E}[\|X\|^2]}}{\min_{X \neq 0}\|X\|}\right). \tag{30}$$

*Proof.* We first prove Eq. 29 by contradiction. Assume the contrary of Eq. 29, we have

$$\begin{aligned}
\mathbb{E}_{Y,X}[\max\{\alpha|\langle Y, X \rangle| - \max\|X\|, 0\}] &\geq \mathbb{E}_Y\big[\max\{\mathbb{E}_X[\alpha|\langle Y, \mathbb{E}[X]\rangle|] - \max\|X\|, 0\}\big] \\
&= \mathbb{E}_Y[\max\{\alpha|\langle Y, \mathbb{E}[X]\rangle| - \max\|X\|, 0\}] \\
&\geq \mathbb{P}_Y[\alpha\mathbb{E}|\langle Y, X \rangle| \geq 2\max\|X\|] \cdot \max\|X\| \\
&> \alpha p(\alpha)\max\|Y\|\max\|X\|.
\end{aligned}$$

However, by Eq. 28, we have

$$\begin{aligned}
\mathbb{E}_{Y,X}[\max\{\alpha|\langle Y, X \rangle| - \max\|X\|, 0\}] &\leq \mathbb{E}_{Y,X}[\max\{\alpha|\langle Y, X \rangle| - \|X\|, 0\}] \\
&\leq \mathbb{E}_X\mathbb{P}_Y[\alpha|\langle Y, X \rangle| \geq \|X\|] \cdot \alpha|\langle Y, X \rangle| \\
&\leq \alpha p(\alpha)\max\|Y\|\max\|X\|,
\end{aligned}$$

contradiction.

Next, we prove Eq. 30 by applying Eq. 29. Denote $\zeta := \min_{X \neq 0} \|X\|$. For any $\xi > 0$, let $k = \left\lceil \log\left( \frac{\sqrt{\mathbb{E}[\|X\|^2]}}{\zeta\sqrt{\xi}} \right) \right\rceil$ and define $a_j := \zeta 2^j$ for each $j \in [k]$. We then define

$$X_j := X \cdot \mathbb{I}\{a_{j-1} \leq \|X\| < a_j\},$$

and

$$X_{k+1} := X \cdot \mathbb{I}\{\|X\| \geq a_k\}$$

which leads to

$$\mathbb{E}\|X\| = \sum_{j=1}^{k+1} \mathbb{E}\|X_j\| \geq \sum_{j=1}^{k} \mathbb{E}\|X_j\|$$

and

$$\alpha \mathbb{E}|\langle Y, X \rangle| = \alpha \sum_{j=1}^{k+1} \mathbb{E}|\langle Y, X_j \rangle| \leq \alpha \sum_{j=1}^{k} \mathbb{E}|\langle Y, X_j \rangle| + \alpha \max \|Y\| \sqrt{\xi \mathbb{E}[\|X\|^2]} \qquad (31)$$

given that

$$\mathbb{E}|\langle Y, X_{k+1} \rangle| \leq \max \|Y\| \cdot \mathbb{E}\|X_{k+1}\|$$

$$\leq \max \|Y\| \cdot \frac{\mathbb{E}[X_{k+1}^2]}{a_k} \leq \max \|Y\| \cdot \frac{\mathbb{E}[\|X\|^2]}{a_k} \leq \max \|Y\| \sqrt{\xi \mathbb{E}[\|X\|^2]}.$$

For any $j = 1, \ldots, k$, we define a new random variable $\widetilde{X}_j$ that satisfies

$$X_j = \begin{cases} \widetilde{X}_j, & \text{w.p. } \mathbb{P}[X_j \neq 0], \\ 0, & \text{w.p. } \mathbb{P}[X_j = 0]. \end{cases}$$

Then,

$$\mathbb{E}\|X_j\| = \mathbb{P}[X_j \neq 0] \cdot \mathbb{E}\|\widetilde{X}_j\| \geq \frac{\mathbb{P}[X_j \neq 0] \cdot \max \|\widetilde{X}_j\|}{2}, \qquad \mathbb{E}|\langle Y, X_j \rangle| = \mathbb{P}[X_j \neq 0] \cdot \mathbb{E}|\langle Y, \widetilde{X}_j \rangle|.$$

By Eq. 29, we have

$$\mathbb{P}_Y[\alpha \mathbb{E}|\langle Y, \widetilde{X}_j \rangle| \geq 2 \max \|\widetilde{X}_j\|] \leq \alpha p(\alpha) \max \|Y\|,$$

which leads to

$$\mathbb{P}_Y[\alpha \mathbb{E}|\langle Y, X_j \rangle| \geq 4 \mathbb{E}\|X_j\|] \leq \alpha p(\alpha) \max \|Y\|$$

and

$$\mathbb{P}_Y\left[ \alpha \sum_{j=1}^{k} \mathbb{E}|\langle Y, X_j \rangle| \geq 4 \sum_{j=1}^{k} \mathbb{E}\|X_j\| \right] \leq k\alpha p(\alpha) \max \|Y\|$$

by union bound. Combining Eq. 31, we can conclude that

$$\mathbb{P}_Y\left[ \alpha \mathbb{E}|\langle Y, X \rangle| > 4 \mathbb{E}\|X\| + \alpha \max \|Y\| \sqrt{\xi \mathbb{E}[\|X\|^2]} \right]$$

$$\leq \alpha p(\alpha) \max \|Y\| \log\left( \frac{\mathbb{E}[\|X\|^2]}{\zeta^2 \xi} \right).$$

Set

$$\xi = \frac{4(\mathbb{E}\|X\|)^2}{\alpha^2 (\max \|Y\|)^2 \mathbb{E}[\|X\|^2]},$$

we obtain

$$\mathbb{P}_Y\left[ \alpha \mathbb{E}|\langle Y, X \rangle| > 8 \mathbb{E}\|X\| \right] \leq \alpha p(\alpha) \max \|Y\| \log\left( \frac{\alpha \max \|Y\| \sqrt{\mathbb{E}[\|X\|^2]}}{\min_{X \neq 0} \|X\|} \right).$$

Since the above inequality holds for any $\alpha \geq 0$, we can rescale $\alpha$ by a factor of 8 and conclude that

$$\mathbb{P}_Y\left[ \alpha \mathbb{E}|\langle Y, X \rangle| > \mathbb{E}\|X\| \right] \leq 8\alpha p(8\alpha) \max \|Y\| \log\left( \frac{\alpha \max \|Y\| \sqrt{\mathbb{E}[\|X\|^2]}}{\min_{X \neq 0} \|X\|} \right).$$

$\square$

**Corollary 9.** *Let $\alpha > 0$. For any random variable $X \in \mathbb{R}^d$ with $\|X\| \leq 1$ and $\min_{X \neq 0} \|X\| \geq \epsilon$, we have*

$$\mathbb{P}_{g \sim G_m^d} \left[ \alpha \mathbb{E} |\langle g, X \rangle| > \max \|X\| \right] \leq 4\alpha\sqrt{d}e^{-1/(2\alpha^2)}$$

*and*

$$\mathbb{P}_{g \sim G_m^d} \left[ \alpha \mathbb{E} |\langle g, X \rangle| > \mathbb{E}\|X\| \right] \leq 16\alpha\sqrt{d}e^{-1/(32\alpha^2)} \log\left(\alpha\sqrt{d}/\epsilon\right).$$

*Proof.* The results are obtained by combining Lemma 19 and Lemma 20. $\qquad \square$

The following lemma establishes that the sum of independent zero-mean sub-Gaussian random variables also follows a sub-Gaussian distribution.

**Lemma 21** (Proposition 2.6.1 of [61]). *For any $k$ independent zero-mean sub-Gaussian random variables $Y_1, \ldots, Y_k$ with variances $\sigma_1^2, \ldots, \sigma_k^2$, their sum $\sum_j^k Y_j$ is also sub-Gaussian with variance $8 \sum_j^k \sigma_j^2$.*

**Lemma 22.** *For any $\epsilon, \delta > 0$, $u \in \mathbb{R}^k$, and $k$ independent random variables $Y_1, \ldots, Y_k$ satisfying*

$$\mathbb{P}\left[ |Y_j - \mathbb{E}[Y_j]| \geq \epsilon \right] \leq \delta, \quad \forall j \in [k], \tag{32}$$

*we have*

$$\mathbb{P}\left[ \left| \sum_{j=1}^k u_j Y_j - \sum_{j=1}^k u_j \mathbb{E}[Y_j] \right| \geq C\|u\|\epsilon \log(1/\delta) \right] \leq 2\delta k. \tag{33}$$

*Proof.* Denote $Z_j := Y_j - \mathbb{E}[Y_j]$. By Eq. 32, each random variable $Z_j$ follows a probability distribution $p_j$ that is $\delta$-close to a probability distribution $\tilde{p}_j$ such that $\max_{Z_j \sim \tilde{p}_j} \|Z_j\| \leq 2\epsilon \log(1/\delta)$. Then, the random variable $Z_j \sim \tilde{p}_j$ follows a sub-Gaussian distribution with variance at most $\epsilon^2$, and the random variable

$$\sum_{j=1}^k u_j Z_j, \quad \{Z_1, \ldots, Z_k\} \sim \tilde{p}_1 \otimes \cdots \otimes \tilde{p}_k, \tag{34}$$

is also a sub-Gaussian distribution with variance at most

$$C^2 \sum_{j=1}^k u_j^2 \mathbb{E}_{Z_j \sim \tilde{p}_j} \|Z_j\|^2 \leq C^2 \|u\|^2 \epsilon^2, \tag{35}$$

where $C$ is the absolute constant in Lemma 21, which leads to

$$\mathbb{P}_{\tilde{p}_1 \otimes \cdots \otimes \tilde{p}_k} \left[ \left| \sum_{j=1}^k u_j Z_j \right| \geq C\|u\|\epsilon \log(1/\delta) \right] \leq \delta. \tag{36}$$

Counting in the difference between $\tilde{p}_j$ and the actual distribution $p_j$, we have

$$\mathbb{P}_{p_1 \otimes \cdots \otimes p_k} \left[ \left| \sum_{j=1}^k u_j Z_j \right| \geq 8\|u\|\epsilon \log(1/\delta) \right] \leq \delta + \delta k = 2\delta k. \tag{37}$$

$$\square$$

