# OpenReview forum: "Isotropic Noise in Stochastic and Quantum Convex Optimization"
_NeurIPS.cc/2025/Conference — NeurIPS 2025 poster_

### Official Review · Reviewer_QeNG · 2025-07-02

**Clarity:** 3
**Significance:** 3
**Originality:** 3
**Rating:** 5
**Confidence:** 4

**Summary:**

This paper studies the stochastic convex optimization (SCO) problem using a stochastic gradient oracle. Under a new assumption that the stochastic gradient is isotropic, the authors develop a quantum algorithm that achieves a speedup of a $d$ factor in certain parameter regimes. This result can be generalized to a stochastic gradient with a sub-exponential noise, therefore achieving the current STOA complexity for stochastic convex optimization with sub-exponential noise. In both cases, the authors also prove matching lower bounds (up to polylogarithmic factors). Therefore, the upper bounds can be regarded as tight. This work can be regarded as a follow-up work of [50] and [64].

**Questions:**

- The isotropic noise seems to be a technical requirement in the proof. Except for sub-Gaussian noise, is there any practical model that exhibits this type of noise distribution?
- Lemma 10 suggested that a $(O(\sqrt{d}), \delta)$-ISGO can be implemented using very few queries to a VSGO. Is it possible to implement a $(O(1), \delta)$ ISGO with $poly(d)$ queries to VSGO (without using quantum)? If so, how would it be worse than Theorem 6?
- Can you comment on other potential computational problems that would benefit from the quantum isotropifier? Or does this technique heavily rely on the assumption of isotropic stochastic noise?

**Ethical Concerns:**

["NO or VERY MINOR ethics concerns only"]

**Limitations:**

Yes

**Paper Formatting Concerns:**

I do not notice major formatting issues in this paper.

**Quality:**

3

**Strengths And Weaknesses:**

Strengths:
- Under the new assumption of isotropic stochastic gradient, an improved query complexity is achieved.
- Matching lower bounds are proven, showing that the proposed quantum algorithms are near-optimal.
- The authors systematically develop a *quantum isotropifier*, which is the key ingredient in the improved bound for quantum SCO.

Weaknesses:
- The improved query complexity results are obtained under stronger theoretical assumptions, such as isotropy and sub-exponential tails.
- The quantum speedup yields only a $d$-factor improvement, which is relatively modest. Considering that the proposed algorithm would require a large fault-tolerant quantum computer, the overhead from quantum error correction could significantly limit its practical applicability.

---

> ### Author Rebuttal · Authors · 2025-07-31
>
> Thank you for your feedback and questions on our paper.
>
> To respond to the weaknesses you raised:
>
> - *"The improved query complexity results are obtained under stronger theoretical assumptions, such as isotropy and sub-exponential tails."*
>
> Though we do require these assumptions for our non-quantum results, we want to emphasize that in our main application to quantum stochastic convex optimization, the isotropic noise model is not an assumption. For quantum stochastic convex optimization, we only assume access to a quantum analog of a VSGO (Definition 6), whose stochastic noise is not guaranteed to be isotropic. In other words, the isotropic noise model is a technical tool which we use to achieve improved rates in a quantum noise model which has been studied previously in the literature.
>
> - *"The quantum speedup yields only a $d$-factor improvement, which is relatively modest..."*
>
> Regarding your comment that our quantum speedup is relatively modest and comes with an additional $d$ factor, we agree that this may limit the applicability of our algorithm on near-term quantum devices. Nevertheless, we would like to note that, as shown in prior work [1], achieving a quantum speedup in $\epsilon$ without incurring additional $d$-dependence is impossible in the worst case. We believe it is an interesting and natural open problem to explore whether our techniques can lead to more practical quantum algorithms given additional assumptions on the stochastic optimization problem.
>
> [1]. Aaron Sidford and Chenyi Zhang. Quantum speedups for stochastic optimization. NeurIPS 2023
>
> To respond to your questions:
>
> - *"The isotropic noise model seems to be a technical requirement in the proof. Except for sub-Gaussian noise, is there any practical model that exhibits this type of noise distribution?"*
>
> Just to emphasize the point we made above, for our main application to quantum stochastic convex optimization, the isotropic noise model is a technical tool as opposed to an assumption.
>
> Regarding non-quantum applications beyond sub-Gaussian distributions, that is a great question. We believe isotropic noise constitutes a natural class of distributions where you can achieve improved rates. We leave it to future work to determine whether actual data distributions satisfy this condition, and whether there are additional classes of problems which can be solved more efficiently in this noise model.
>
> - *"Is it possible to implement a  $(O(1), \delta)$ ISGO with $\mathrm{poly}(d)$ queries to VSGO (without using quantum)? If so, how would it be worse than Theorem 6?"*
>
> Regarding your question about implementing an $(O(1), \delta)$-ISGO with $\text{poly}(d)$ queries to a VSGO: Using a sub-Gaussian mean estimator (e.g., Theorem 8 in [2]), a $(O(1), \delta)$-ISGO can be implemented with $d$ queries (up to log factors) to a VSGO, whereas Theorem 6 is able to achieve $\sqrt{d}$ queries (up to log factors).
>
> [2]. Lugosi and Mendelson. Mean Estimation and Regression Under Heavy-Tailed Distributions: A Survey. Foundations of Computational Mathematics.
>
> - *“Can you comment on other potential computational problems that would benefit from the quantum isotropifier?”*
>
> We think that is a great question. We are only using it inside cutting plane methods for now, but it is an interesting question if other problems, particularly those related to the applications of quantum multivariate mean estimation, might benefit or if it affects practical performance. However, we leave this for future work.
>
> - *“Does this technique heavily rely on the assumption of isotropic stochastic noise?”*
>
> Isotropic noise is not an assumption for the technique. Rather, it produces isotropic noise from a (potentially) non-isotropic distribution.

---

### Official Review · Reviewer_62sv · 2025-07-02

**Clarity:** 3
**Significance:** 2
**Originality:** 3
**Rating:** 4
**Confidence:** 3

**Summary:**

This work introduces a setting where the noise of the stochastic gradient is bounded in every direction with high probability, i.e., the noise is isotropic. As a special case, the proposed algorithm achieves the best complexity which improves a factor of $d$ in certain regimes, for sub-exponential noise. Matching lower bounds are also provided under this noise model.

**Questions:**

- Why is $R^2 \sigma_V^2 / \epsilon^2$ unimprovable when $d=1$? Isn't the optimality with $\sigma_B^2$ instead?

- What is the query complexity of SGD under the same noise model (ISGO)?

- Can the authors comment on the sensibility of the isotropic noise model? Is it the same model as what is being referred in [1]?

- It seems that there exists dimension independent quantum algorithms for similar problem setting. How does this work compare with [2] for example?

[1] H. Daneshmand et al., "Escaping Saddles with Stochastic Gradients"

[2] B. Augustino et al., "Fast Convex Optimization with Quantum Gradient Methods"

**Ethical Concerns:**

["NO or VERY MINOR ethics concerns only"]

**Final Justification:**

The authors have provided a fairly detailed reply, which addresses my previous concerns. I think the proposed regime is still quite specific, but I recommend acceptance as achieving SOTA complexity for subexponential noise, having a lower bound, and proposing a quantum isotropifier are valuable to the community.

**Limitations:**

Yes

**Paper Formatting Concerns:**

No major formatting issue.

**Quality:**

3

**Strengths And Weaknesses:**

**Strength:**
- Achieves the best known complexity for sub-exponential noise, which is $\tilde{O} (R^2 \sigma^2 / \epsilon^2 + d)$ query complexity.

- The quantum isotropifier, which converts a variance bounded quantum sampling oracle to an unbiased estimate with isotropic error can be of an independent interest.


**Weakness:**
- The considered regime is fairly specific, and the posed open problem is not completely resolved (other than sub-exponential noise).
- It would be great if the authors can comment on the sensibility of the noise model. It feels that the noise model was introduced in order to justify for the quantum isotropifier. From the stochastic convex optimization perspective, though, it is unclear if the assumption is reasonable.

---

> ### Author Rebuttal · Authors · 2025-07-31
>
> Thank you for your feedback and questions on our paper.
>
> To respond to the weaknesses you raised:
>
> - *"The posed open problem is not completely resolved (other than sub-exponential noise)."*
>
> In regard to open problems, key quantitative contributions of this paper are tight rates for ISGOs and faster quantum stochastic convex optimization. Indeed, we do present what we believe is a natural further open problem suggested by this line or work, but we kindly suggest reviewers view this as an additional open problem or contribution, which is intended to encourage the community to further improve upon our results (rather than a failing of them). Also, note that resolving the open problem wouldn’t immediately yield faster quantum stochastic convex optimization algorithms than our approach, due to our quantum isotropifier. We hope that our work may facilitate further research on this fundamental open problem that we highlight.
>
> - *"It would be great if the authors can comment on the sensibility of the noise model. It feels that the noise model was introduced in order to justify the quantum isotropifier."*
>
> We want to emphasize that in our main application to quantum stochastic convex optimization, the isotropic noise model is not an assumption. For quantum stochastic convex optimization, we only assume access to a quantum analog of a VSGO (Definition 6), whose stochastic noise is not guaranteed to be isotropic. In other words, the isotropic noise model is a technical tool which we use to achieve improved rates in a quantum noise model which has been studied previously in the literature.
>
> Regarding the sensibility/applicability of the noise model beyond quantum stochastic convex optimization, that is a great question. We identify what we believe is a natural class of distributions where you can achieve improved rates. We leave it to future work to determine whether actual data distributions satisfy this condition, and whether there are additional classes of problems which can be solved more efficiently in this noise model.
>
> To respond to your questions:
>
> - *"Why is $R^2 \sigma_V^2 / \epsilon^2$ unimprovable when $d = 1$?"*
>
> $R^2 \sigma_V^2 / \epsilon^2$ can be shown to be unimprovable when $d = 1$ using the same lower bound construction which shows that $R^2 \sigma_B^2 / \epsilon^2$ is unimprovable when $d = 1$. Indeed, in that construction (e.g., Section 5 in [1]) it is the case that $\sigma_V^2 = \Theta(\sigma_B^2)$. We will include this point in our revision.
>
> [1]. John Duchi. Introductory Lectures on Stochastic Optimization. IAS/Park City Mathematics Series, American Mathematical Society.
>
> - *"What is the query complexity of SGD under the same noise model (ISGO)."*
>
> Regarding the query complexity of SGD under an ISGO: When $\delta$ is sufficiently small as an inverse polynomial in the problem parameters (analogously to Theorem 1), then SGD achieves an $\tilde{O} ( R^2 (L^2 + \sigma_I^2) / \epsilon^2)$ rate with high probability. In this case, a $(\sigma_I, \delta)$-ISGO query implements a $O(L + \sigma_I)$-BSGO (Definition 2) query with high probability, in which case we can apply the classical SGD rate for BSGOs.
>
> - *"Can the authors comment on the sensibility of the isotropic noise model?"*
>
> See above.
>
> - *"Is it the same model as what is being referred to in [1]."*
>
> We do not believe it is the same model. I could not find a formal definition of “isotropic noise'' in that paper or the papers it cites under the “isotropic noise'' header on the first page. To my understanding, when that paper refers to isotropic noise they mean noise which must have a component in every direction (see "exhibits a certain amount of variance along all directions in $\mathbb{R}^d$" on page 1), namely a lower bound on how much noise the distribution has in each direction, whereas our definition corresponds to an upper bound on how much noise the distribution has in each direction.
>
> - *"It seems that there exists dimension independent quantum algorithms for similar problem setting. How does this work compare with [2] for example?"*
>
> Thank you for your question regarding the comparison between our work and that of B. Augustino et al. We would like to clarify that these works belong to two distinct lines of research on quantum algorithms for continuous optimization:
> The first line, which includes [2, 3] and the work by Augustino et al., focuses on algorithms that access a (possibly approximate) quantum value oracle. In this setting, the oracle’s output need not be unbiased but must have small errors. For instance, in Theorem 2.1 of Augustino et al., the required error is of the order $\mathrm{poly}(\epsilon)/\mathrm{poly}(d)$.
>
> The second line, which includes [4] and our work, considers access to a quantum stochastic gradient oracle. In this setting, the oracle provides unbiased gradient estimates with variance that is independent of both $\epsilon$ and $d$. Notably, [4] proves that a dimension-independent quantum speedup is impossible in this setting.
>
> We thank you for pointing out the relevant work by Augustino et al. and may include some discussion of this work in our final version.
>
> [2]. Joran van Apeldoorn, András Gilyén, Sander Gribling, and Ronald de Wolf. Convex optimization using quantum oracles. Quantum 2020
>
> [3]. Shouvanik Chakrabarti, Andrew M. Childs, Tongyang Li, and Xiaodi Wu. Quantum algorithms and lower bounds for convex optimization. Quantum 2020
>
> [4]. Aaron Sidford and Chenyi Zhang. Quantum speedups for stochastic optimization. NeurIPS 2023

---

> > ### Comment · Reviewer_62sv · 2025-08-06
> >
> > Dear authors,
> >
> > Thank you for the detailed reply. My previous concerns are all addressed in the reply, and I have updated the score accordingly.

---

### Official Review · Reviewer_VDzC · 2025-07-03

**Clarity:** 3
**Significance:** 3
**Originality:** 3
**Rating:** 4
**Confidence:** 4

**Summary:**

This paper investigates the problem of stochastic convex optimization under a noise model termed isotropic noise, which includes white additive gaussian model as a special case, and proposes algorithms that improves the state of the art sample complexities. The contribution is two fold. First, in the classical setting with first-order oracle, the authors propose a cutting-plane-based optimization algorithm that achieves an upper bound of $\tilde{O}(R^2\sigma^2/\epsilon^2+d)$ queries under the isotropic noise model for convex and Lipschitz functions. This improves the standard SGD in certain regimes, which has a sample complexity of ${O}(R^2\sigma^2/\epsilon^2+R^2L^2/\epsilon^2)$. A matching lower bound is provided to show that the better of the two algorithms is sample optimal up to logarithmic factors. Second, the authors extended their results to a quantum setting, where $1/\epsilon$ complexity can be achieved through the well known amplitude amplification approach. This extended algorithm improves previous results in the quantum domain by a polynomial factor in dimension $d$.

**Questions:**

While the reviewer has an overall positive view on the presented results. The current manuscript has a few issues to be resolved. The reviewer would be happy to raise the score if they are addressed adequately.

1. As discussed above in the weakness section, since the quantum oracle considered in this paper provides access to additional information of the objective function compared to the classical case, which makes it an algorithmically easier problem. The reviewer recommends the authors to carefully revise relevant statements such as referring the oracle as "quantum generalizations of classical oracles" and the $1/\epsilon$ complexity as "quantum speedups".

2. In terms of the motivation of the quantum oracle formulation, the authors provided a supporting argument in the footnote on page 4 from the aspect of oracle implementation, mentioning that such quantum oracle can be implemented if the implementation of the classical oracle is available. However, in optimization problems the identity of the circuit for the classical oracle can not be known, otherwise we always have a sample complexity of O(1). Could the authors clarify how such quantum oracle can be constructed without accessing the circuit of classical oracle?

3. Could the authors provide additional motivation on the particular quantum oracle formulation in this work besides that it has been considered in prior work?

4. Do the authors have any conjectures or insights regarding lower bounds in the quantum setting?

**Ethical Concerns:**

["NO or VERY MINOR ethics concerns only"]

**Final Justification:**

My review comments are addressed in the second response of the authors, where they provided a high-level plan for the revision. While the exact revision is not provided, the issues raised in the review might be resolved if the manuscript is revised appropriately.

Given the uncertainty and the partial resolution of the concerns, my score remains unchanged. However, I do not strongly oppose the acceptance of this paper.

**Limitations:**

yes

**Quality:**

3

**Strengths And Weaknesses:**

Strengths:
The technical results are sound, with detailed proofs provided in the appendix. To the best of the reviewer's knowledge, the presented achievability of the optimal sample complexity in the classical case is new. Given that finding optimal sample complexities for Lipschitz convex functions is a fundamental problem, the achievability result presented in this paper that matches the lower bound is quite significant, albeit requiring the isotropic noise assumption.

Weakness:
1. The clarity and structure of writing can be improved, particularly for the introduction. Especially, it would be helpful to state the assumptions on the objective functions upfront, i.e., Lipschitz and convex, since the introduction explores various frameworks including ones with Lipschitz gradient. Stating the assumptions clearly helps readers better understand the strength of the proposed result. Besides, it is not clear how the results for the Lipschitz gradient case is connected to the results in the framework in this paper. The authors are recommended to either clarify their connections, or to move those discussions to a separate paragraph at the end of introduction as overview of related works.

2. A significant limitation of this work is on the quantum formulation, where the authors assumed the quantum oracle essentially provides a pure-state outcome, specifically, the garbage state that is entangled to the gradient outcome $|g>$ is assumed to be known and used in the amplitude amplification part of the algorithm. While this is a typical assumption in related works, it is technically incorrect to claim that this formulation is a generalization to the classical case and the proposed algorithm achieves a "quantum speedup", as in the classical case the garbage states are viewed as part of the environment of which the identity is unknown and is not accessible by the algorithm. The reviewer believes that if we consider the most general non-pure state quantum oracles then the $1/\epsilon$ complexity is not achievable and the optimal sample complexity would reduce to the result in the classical case.

3. While the bounds under isotropic noise and sub-exponential noise are tight up to log factors, the main open problem of achieving optimal sample complexities for variance-bounded noise remains unresolved.

---

> ### Author Rebuttal · Authors · 2025-07-31
>
> Thank you for your feedback and questions on our paper.
>
> To respond to the weaknesses you raised:
>
> - *"It would be helpful to state the assumption on the objective functions upfront, i.e., Lipschitz and convex."*
>
> Note that we state that $f$ is convex and $L$-Lipschitz in Lines 19 - 22. More broadly, the introduction was structured with the intent of making the assumptions clear. If you could identify places in the paper where you feel we missed a statement or an explanation, please let us know; we would be grateful and are eager to further improve the paper’s clarity.
>
> - *"It is not clear how the results for the Lipschitz gradient case is connected to the results in the framework in this paper."*
>
> Thank you for this question. We do mention prior dimension-independent work on the Lipschitz gradient case in Line 37 when explaining the prior work on SCO with a VSGO oracle. Additionally, our bounds can be extended immediately to the Lipschitz gradient case at the cost of only logarithmic factors since our bounds have logarithmic dependence on the Lipschitz constant of $f$. This follows because a function with a $G$-Lipschitz gradient which is minimized in a ball of radius $R$ is itself $O(GR)$-Lipschitz in the ball. We will mention this in our revision.
>
> - *“A significant limitation of this work is on the quantum formulation, where the authors assumed the quantum oracle essentially provides a pure-state outcome, specifically, the garbage state that is entangled to the gradient outcome $|g>$ is assumed to be known and used in the amplitude amplification part of the algorithm.”*
>
> Thank you for this question. We would like to clarify that the garbage state is not assumed to be known by our algorithm. This is because quantum amplitude amplification requires only the ability to apply the oracle and its inverse as a black box, without knowledge of the garbage state. Accordingly, our algorithm only needs to maintain coherence with the environment but does not require knowledge of its identity.
>
> - *“While this is a typical assumption in related works, it is technically incorrect to claim that this formulation is a generalization to the classical case and the proposed algorithm achieves a "quantum speedup", as in the classical case the garbage states are viewed as part of the environment of which the identity is unknown and is not accessible by the algorithm.”*
>
> Thank you for raising the question regarding our use of the term “quantum speedups.” We adopted this terminology in part because it is commonly used in the literature (e.g., [1, 2]), and also due to the widespread convention of referring to Grover’s algorithm as achieving a “quadratic speedup.” That said, we can see how this phrasing may lead to confusion, and will look into alternative expressions in the final version.
>
> - *"While the bounds under isotropic and sub-exponential noise are tight up to log factors, the main open problem of achieving optimal sample complexities for variance-bounded noise remains unresolved."*
>
> In regard to open problems, key quantitative contributions of this paper are tight rates for ISGOs and faster quantum stochastic convex optimization. Indeed, we do present what we believe is a natural further open problem suggested by this line of work, but we kindly suggest reviewers view this as an additional open problem or contribution, which is intended to encourage the community to further improve upon our results (rather than a failing of them). Also, note that resolving the open problem wouldn’t immediately yield faster quantum stochastic convex optimization algorithms than our approach, due to our quantum isotropifier. We hope that our work may facilitate further research on this fundamental open problem that we highlight.
>
> To respond to your questions:
>
> - We thank you for raising the question regarding our use of the term “generalization” for our quantum oracle. We use this term to convey that the quantum stochastic gradient oracle is strictly more powerful than its classical counterpart, since the latter can be obtained by measuring the quantum oracle. Given known query complexity lower bounds for algorithms using classical stochastic oracles, improved rates are only possible with access to strictly more powerful oracles. We acknowledge that the term “generalization” may cause confusion in this context, and will fix this in our final version.
>
> - Thank you for your question regarding the implementation and motivation of the quantum oracle. We would like to clarify that, in the worst case, we do not see how to implement the quantum oracle efficiently using a classical oracle without incurring a loss in query complexity. We did not intend to suggest otherwise, and if any part of the paper leads to this confusion, we are happy to try to revise it for clarity.
>
> - Regarding your question on the access to the identity for the circuit to the classical oracle, indeed, knowing the gradient circuit makes the query complexity of the problem trivial. However, it doesn’t necessarily make the problem computationally trivial. As discussed in [3, 4], even if the classical oracle’s internal circuit is known, computing the gradient exactly from this description may require exponential time in the circuit size (as far as we know). Therefore, it remains meaningful to study algorithms that access the oracle in a black-box manner, which is the motivation behind the transformation we described. If there are specific parts of the paper that would benefit from further clarification, we would be more than happy to expand on them in the final version.
>
> - Regarding the motivation of the quantum oracle, we are not aware of any key motivation for considering this setting that was not already present in prior works. For this reason, we simply cited the prior work rather than repeat these motivations.
>
> - Regarding your question on lower bounds in the quantum setting, we agree that this is a good question. We would like to note that a lower bound is established in prior work [2]. However, this lower bound does not match our algorithmic result, and it is unclear to us how to improve the existing construction. We may include this as an open question in the revised version.
>
> [1]. Ashley Montanaro. Quantum speedup of Monte Carlo methods. Proceedings of the Royal Society A: Mathematical, Physical and Engineering Sciences 2015.
>
> [2]. Aaron Sidford and Chenyi Zhang. Quantum speedups for stochastic optimization. NeurIPS 2023
>
> [3]. Robin Kothari, and Ryan O’Donnell. Mean estimation when you have the source code; or, quantum Monte Carlo methods. SODA 2023.
>
> [4]. Clément L. Canonne, Robin Kothari, and Ryan O'Donnell. Uniformity testing when you have the source code. 2024.

---

> > ### Comment · Reviewer_VDzC · 2025-08-05
> >
> > I have read the authors’ response. While several points were addressed, many key concerns and suggestions were not directly engaged with, seemingly due to misinterpretations of the reviewer's questions or underlying formulations, particularly regarding the connection between quantum and classical systems. We remind the authors that the review process is intended to support constructive revision, rather than defend potentially unclear or misleading elements of the current manuscript.
> >
> > Below we clarify some of the outstanding concerns:
> >
> > 1. The suggestion regarding Lipschitz and convex was to clearly state them as assumptions upfront, rather than one of the examples. The authors' response points to Lines 19–22 as evidence. However, that part of the manuscript precisely presents these properties only as examples within a broader discussion and not as formal assumptions that underlie the results. If the rest of the paper does indeed rely on such assumptions, they should be clearly stated in the main definitions or problem statements (e.g., Definitions 1, 2, or Open Problem 1). Presenting assumptions as examples can lead to confusion about the actual scope of the results.
> >
> > 2. The reviewer suggested separating the discussion of the alternative setting of Lipschitz gradient into a related work section, as its connection to the main formulation was unclear. The response first reiterates its presence in the introduction, which does not address the suggestion nor clarify the relationship between the two frameworks or plans for restructuring. Following this statement, the authors' response includes a brief discussion on the relationship between the two formulations, which appears more as an extension than a component essential to the main setup, and thus would still be more appropriate in a separate section. However, we expected a concrete plan for how this discussion in the manuscript will be revised and relocated.
> >
> > 3. The response asserts that amplitude amplification only requires oracle access and its inverse as black boxes, and thus "does not require knowledge of the garbage state." This is technically not true. Open quantum systems are not invertible and the fact that an inverse oracle is assumed utilizes the knowledge of the garbage state.
> >
> > 4. The authors maintain the use of “quantum speedup,” referencing Grover’s algorithm and some other related works. Firstly, the claim of speedup requires a well-defined and realizable oracle implementation, which is true in the setting of Grover’s
> >  search algorithm. As elaborated in the earlier review this is not the case in the optimization setting considered here. The authors' response indeed acknowledges that no concrete quantum oracle is proposed or implemented in the paper, which raises questions about the validity of the speedup claim. If related literature has used this terminology loosely, that should be treated as a motivation for required further clarity in the manuscript rather than justification for maintaining potentially misleading terminology.
> >
> > In summary, I encourage the authors to revisit the manuscript with a focus on improving clarity and precision, particularly in stating assumptions and avoid overclaims, organizing the presentation of alternative formulations, and framing claims with appropriate rigor. These revisions would strengthen the paper and help ensure that readers can interpret its contributions correctly and consistently with accepted standards of scientific communication.

---

> > > ### Author Response · Authors · 2025-08-06
> > >
> > > Thank you to the reviewer for their continued feedback, efforts to make sure that points are not missed, and work to clear up potential sources of confusion. It was our intent to take all of your earlier feedback into consideration both in our response and in revising our submission; we apologize if our review seemed to convey otherwise. We felt it important to explain what was in our submission and how we perceived its relationship to your earlier comments in part to mitigate potential points of confusion and to help ensure that our revisions ultimately improve the quality of the paper. We hope that the multiple mentions to our plans to revise our manuscript further clarify our intent.
> > >
> > > In regard to the specific outstanding concerns, thank you again for raising them and for your efforts to benefit the manuscript. In response:
> > >
> > > 1\) We plan to add additional text to the first two pages to further ensure it is clear from the onset that we focus on the setting where $f$ is convex and $L$-Lipschitz, as well as additional pointers to the contents of Section 1.1, where we formally state our full results with self-contained assumptions.
> > >
> > > More broadly, while we are eager to better connect the motivating text given in the first two pages of our paper to the formal statement of our results, we want to emphasize that the latter is given in Section 1.1, starting at the bottom of the second page. Indeed, we believe the formal statement of our problem setting (Problem 1 on page 3) as well as our main theorems (Theorem 1, Corollaries 2, 3, etc.) are fully rigorous with clear, self-contained assumptions. If the reviewer sees otherwise, please let us know. We believe this template of first giving motivation (the primary focus of our first two pages) followed directly by a rigorous statement of our results (Section 1.1) is standard for NeurIPS. However, we hope the planned changes we outlined above to the motivating text on the first two pages will further mitigate potential confusion at the onset.
> > >
> > > 2\) Thank you for this feedback. After additional discussion, we agree that bringing up the Lipschitz-gradient case in the first two pages, when it is not the setting we focus on, is unnecessary to motivate our results. We plan to move discussion of the Lipschitz-gradient setting to a separate related-work section, where we plan to state what is known about the Lipschitz-gradient setting and discuss how our results also apply to this setting through standard reductions.
> > >
> > > 3\) Regarding your comment on the requirement on the garbage state, we agree that one needs to either know the identity of the garbage state to perform the inverse operation, as in an open quantum system, or maintain the garbage state coherently as part of the system, as in a closed quantum system. In our work, we consider the latter setting, where the garbage state is maintained during the algorithm, as in prior works [1, 2, 3]. We will further clarify this setting and be explicit about the requirement of maintaining the garbage state in the final version.
> > >
> > > 4\) Regarding your comment on our use of the term “quantum speedups,” we agree that this phrasing may lead to potential confusion, and we will use alternative expressions in the final version. We thank you again for bringing this to our attention. Our intention was not to overstate the scope of our result; rather, we used the term in the broader sense that appears in several prior works, including in oracle settings (e.g., [4, 5, 6]). That said, we appreciate the concern that such terminology could be misleading, and we agree that more precise phrasing would help improve clarity.
> > >
> > > Again, thanks to the reviewers extensive efforts to clarify concerns and further improve the manuscript.
> > >
> > > [1]. Ashley Montanaro. Quantum speedup of Monte Carlo methods. Proceedings of the Royal Society A: Mathematical, Physical and Engineering Sciences 2015.
> > >
> > > [2]. Arjan Cornelissen, Yassine Hamoudi, and Sofiene Jerbi. Near-optimal quantum algorithms for multivariate mean estimation. STOC 2022.
> > >
> > > [3]. Aaron Sidford and Chenyi Zhang. Quantum speedups for stochastic optimization. NeurIPS 2023.
> > >
> > > [4]. Andrew M. Childs,, Richard Cleve, Enrico Deotto, Edward Farhi, Sam Gutmann, and Daniel A. Spielman. Exponential algorithmic speedup by a quantum walk. STOC 2003.
> > >
> > > [5]. Shalev Ben-David, Andrew M. Childs, András Gilyén, William Kretschmer, Supartha Podder, and Daochen Wang. Symmetries, graph properties, and quantum speedups. FOCS 2020.
> > >
> > > [6]. Ryan Babbush, Dominic W. Berry, Robin Kothari, Rolando D. Somma, and Nathan Wiebe. Exponential quantum speedup in simulating coupled classical oscillators. Physical Review X.

---

### Official Review · Reviewer_BTYs · 2025-07-03

**Clarity:** 4
**Significance:** 4
**Originality:** 4
**Rating:** 5
**Confidence:** 4

**Summary:**

This submission studies quantum and classical algorithms for convex optimization. In particular, the authors propose isotropic stochastic gradient oracles in the sense that the noise in each direction is bounded with large probability. Using this new oracle, the problems of convex optimization can be solved faster than using SGO in the high precision and low variance regime.

The authors also proposed a quantum bounded stochastic gradient oracle, which can be used to improve the quantum convex optimizer because the classical isotropic stochastic gradient oracle can be efficiently implemented using this quantum oracle, based on quantum unbiased phase estimation.

**Questions:**

Some minor comments:
1. Line 53 seems not grammatically correct.
2. Line 304: missing "is" at the end of the line?
3. Eq. (3): the use of continuous-space states is not properly defined. (See Section 3 of arXiv:1908.03903).

**Ethical Concerns:**

["NO or VERY MINOR ethics concerns only"]

**Final Justification:**

My questions are mostly addressed. I keep the positive score.

**Limitations:**

yes

**Quality:**

4

**Strengths And Weaknesses:**

I feel this submission makes a solid theoretical contribution in classical and quantum optimization. Notably, a lower bound for the classical case is also given.

The major weakness lies in the motivation of the oracle: are there real-time applications of convex optimization that can provide the isotropic oracle without significant overhead?

---

> ### Author Rebuttal · Authors · 2025-07-31
>
> Thank you for your feedback on our paper and positive recognition of our contributions. We will correct the typos that you raised in our revised version.
>
> *“The major weakness lies in the motivation of the oracle: are there real-time applications of convex optimization that can provide the isotropic oracle without significant overhead?”*
>
> We want to emphasize that in our main application to quantum stochastic convex optimization, the isotropic noise model is not an assumption. For quantum stochastic convex optimization, we only assume access to a quantum analog of a VSGO oracle (Definition 6), whose stochastic noise is not guaranteed to be isotropic. In other words, the isotropic noise model is a technical tool which we use to achieve improved rates in a quantum noise model which has been studied previously in the literature.
>
> Regarding non-quantum applications, that is a great question. We identify what we believe is a natural class of distributions where you can achieve improved rates. We leave it to future work to determine whether actual data distributions satisfy this condition, and whether there are additional classes of problems which can be solved more efficiently in this noise model.

---

> > ### Comment · Reviewer_BTYs · 2025-08-08
> >
> > Thanks for your response. My concerns and questions are addressed.

---

### Decision · Program_Chairs · 2025-09-17

**Decision:**

Accept (poster)

**Comment:**

All reviewers strongly agree that this submission makes a valuable contribution to both quantum and classical optimization. At the same time, we urge the authors to take the feedback seriously—particularly with respect to the presentation of the results—and to incorporate the necessary revisions.